# A new hERG allosteric modulator rescues genetic and drug-induced long-QT syndrome phenotypes in cardiomyocytes from isogenic pairs of patient induced pluripotent stem cells

Luca Sala[1], Zhiyi Yu[2], Dorien Ward-van Oostwaard[1], Jacobus PD van Veldhoven[2], Alessandra Moretti[3], Karl-Ludwig Laugwitz[3], Christine L Mummery[1,†], Adriaan P IJzerman[2,†] & Milena Bellin[1,*,†]

## Abstract

Long-QT syndrome (LQTS) is an arrhythmogenic disorder characterised by prolongation of the QT interval in the electrocardiogram, which can lead to sudden cardiac death. Pharmacological treatments are far from optimal for congenital forms of LQTS, while the acquired form, often triggered by drugs that (sometimes inadvertently) target the cardiac hERG channel, is still a challenge in drug development because of cardiotoxicity. Current experimental models *in vitro* fall short in predicting proarrhythmic properties of new drugs in humans. Here, we leveraged a series of isogenically matched, diseased and genetically engineered, human induced pluripotent stem cell-derived cardiomyocytes (hiPSC-CMs) from patients to test a novel hERG allosteric modulator for treating congenital LQTS, drug-induced LQTS or a combination of the two. By slowing $I_{Kr}$ deactivation and positively shifting $I_{Kr}$ inactivation, the small molecule LUF7346 effectively rescued all of these conditions, demonstrating in a human system that allosteric modulation of hERG may be useful as an approach to treat inherited and drug-induced LQTS. Furthermore, our study provides experimental support of the value of isogenic pairs of patient hiPSC-CMs as platforms for testing drug sensitivities and performing safety pharmacology.

**Keywords** cardiac arrhythmia; drug screening; hERG; human induced pluripotent stem cells; long-QT syndrome
**Subject Categories** Cardiovascular System; Stem Cells

## Introduction

Long-QT syndrome (LQTS) is a cardiac disorder primarily characterised by the prolongation of the QT interval on an electrocardiogram. This increases the propensity for life-threatening arrhythmias in a structurally normal heart (Priori *et al*, 2001; Schwartz *et al*, 2012). Both congenital and acquired forms of LQTS have been described. Congenital forms are caused by mutations in one of 16 different genes (Schwartz *et al*, 2013), most patients presenting with mutations in *KCNQ1* [defined as LQT1 when heterozygous and Jervell and Lange-Nielsen syndrome (JLNS) when homozygous and associated with deafness (Jervell & Lange-Nielsen, 1957)], *KCNH2* (LQT2) or *SCN5A* (LQT3) genes (Schwartz *et al*, 2001). The acquired form by contrast is triggered in healthy individuals and LQTS mutation carriers by ancillary causes such as bradycardia, electrolyte abnormalities or drugs that target cardiac ion channels non-specifically (Roden *et al*, 1996; Zareba *et al*, 2003; Itoh *et al*, 2016). The human ether-à-go-go-related gene (hERG) channel (also known as $K_v11.1$) is the most likely to be affected (Saenen & Vrints, 2008; Mahida *et al*, 2013). The concomitant presence of (known or unknown) polymorphisms affecting gene expression levels of either ion channels or enzymes involved in drug metabolism can make patients more susceptible to those stimuli and may facilitate the occurrence of the typical LQTS arrhythmia called "Torsades de Pointes". In some cases, removing the damage-inducing stimulus may be sufficient to treat the acquired form of LQTS. However, pre-existing medical conditions that require (life-saving) drugs can complicate or preclude stopping treatment. Even though current therapies for congenital LQTS generally give fairly good clinical responses, they could still be improved (Schwartz, 2015). Therefore, there remains a need for new pharmacological approaches to prevent sudden cardiac death (Ruan *et al*, 2008). Recent studies investigated the effect of novel hERG activators (Bentzen *et al*, 2011; Zhang *et al*, 2012; Bebernitz *et al*, 2015; Giacomini *et al*,

1  Department of Anatomy and Embryology, Leiden University Medical Center, Leiden, The Netherlands
2  Gorlaeus Laboratories, Leiden Academic Centre for Drug Research, Leiden University, Leiden, The Netherlands
3  I. Department of Medicine (Cardiology), Klinikum rechts der Isar, Technical University of Munich, Munich, Germany
   *Corresponding author. Tel: +31 715269382; Fax: +31 715268289; E-mail: m.bellin@lumc.nl
   †These authors contributed equally to this work

2015; Mannikko *et al*, 2015; Bossu *et al*, 2016; Yu *et al*, 2016), which indicated that there is significant interest from academia and pharma in developing small molecules to increase the rapid component of the delayed rectifying potassium current ($I_{Kr}$) conducted by hERG channel, since this could counteract both congenital and acquired LQTS. Furthermore, pharmacologically targeting and activating hERG could be an improvement on present strategies, since this channel is the most commonly mutated in asymptomatic/borderline carriers with acquired LQTS (Itoh *et al*, 2016). However, the challenge here is to generate experimental models that accurately reproduce all of the clinical features of the disease. Although crucial for basic characterisation of LQTS-causing mutations, neither heterologous systems (Winbo *et al*, 2009; Diamant *et al*, 2013) nor animal models (Salama & London, 2007) fully recreate the proper disease phenotype complexity. This limits the possibility of developing genotype-specific or patient-specific therapies, which already appear to be the way forward in treating congenital cardiac arrhythmias (Priori, 1998; Shimizu *et al*, 2005) and, more recently, in cardio-oncology (Burridge *et al*, 2016).

Human pluripotent stem cell-derived cardiomyocytes [hPSC-CMs, which include cardiomyocytes from human induced pluripotent stem cells (hiPSC-CMs) and human embryonic stem cells (hESC-CMs)] represent a powerful tool for *in vitro* disease modelling as they are able to recreate the phenotypical traits of both monogenic and complex diseases (Moretti *et al*, 2010; Bellin *et al*, 2013; Zhang *et al*, 2014; de Boer & Eggan, 2015; Freedman *et al*, 2015). The possibility of obtaining patient-specific cells has strengthened the link between clinical data and *in vitro* phenotype (Moretti *et al*, 2010). More recently, the generation of genetically matched isogenic pairs demonstrated that the genetic background strongly influences both the pathological phenotype and drug response (Bellin *et al*, 2013; Zhang *et al*, 2014). The combination of these two approaches provides robust disease models with genetically matched controls enabling the recapitulation of clinical features of a specific disorder, in this case LQTS, and at the same time a valuable, multipurpose platform for drug screening in disease and safety pharmacology.

Current Food and Drug Administration (FDA) guidelines recommend *in vitro* hERG assays as a prelude to *in vivo* studies in animals (ICH, 2005). The rationale is that, by assessing the *in vitro* affinity of a molecule for the hERG channel, the risk of arrhythmogenic events *in vivo* can be predicted. However, incidental withdrawal of drugs from the market and the conspicuous number of drugs that do not enter clinical phases of development because of indications for cardiac adverse effects demonstrate that current risk assessment assays are still not completely predictive (Roden, 2005). Patient-specific hiPSC-CMs are now being proposed as a complementary model for safety pharmacology (Braam *et al*, 2010), especially for detecting the proarrhythmic potential of drugs before moving to tests in animals, and as an alternative to the combination of multiple low cost, but poorly predictive, single ion channel assays (Kramer *et al*, 2013).

Here, we tested LUF7346, one of a series of novel hERG allosteric modulators (Yu *et al*, 2015, 2016) on a platform of LQT1, JLNS, LQT2, and control isogenic human pluripotent stem cell pairs to (i) rescue the genetic form of LQTS, (ii) reverse drug-induced LQTS and (iii) correct the combination of genetic and drug-induced LQTS. Application of the small hERG allosteric modulator normalised both action- and field potentials (AP and FP, respectively) in all hPSC-CMs by slowing $I_{Kr}$ deactivation and positively shifting the $I_{Kr}$ inactivation. The compound also normalised the beat-to-beat variability of the repolarisation duration (BVR) and rescued the arrhythmogenic phenotype observed in JLNS-CMs. Our results contribute to validating the use isogenic pairs of hPSC-CMs in drug discovery and safety pharmacology.

# Results

## Identification of hERG allosteric modulators

We recently synthesised a series of small molecules that may act as allosteric modulators of the hERG channel in a heterologous cell system (Yu *et al*, 2015) and in primary rodent cardiomyocytes (Yu *et al*, 2016). The activity of five of these molecules (chemical structures in Fig 1A) was assessed by measuring their effect on the dissociation characteristics of a radioactive probe, [$^3$H]dofetilide, from the hERG channel (Fig 1B). All of the selected compounds, at both 10 and 50 μM, significantly increased the dissociation of [$^3$H]dofetilide from the channel by 29–84% (Appendix Table S1) implying their negative allosteric modulation of the channel inhibition by hERG blockers like dofetilide. Among these compounds, LUF7346 had the most prominent allosteric effect on the hERG channel at 50 μM. Consistently, pilot experiments in hiPSC-CMs using microelectrode arrays (MEA) confirmed a stronger effect of this compound on FP prolongation than all of the other shortlisted chemicals (Appendix Fig S1). We therefore decided to continue with LUF7346 in further experiments.

## LUF7346 increases $I_{Kr}$ in a heterologous system

We first used HEK293 cells stably expressing the hERG channel (HEK293 hERG) as a simple biological system to assess the effect of LUF7346 on $I_{Kr}$ biophysical properties (Masi *et al*, 2005). Figure 1C shows representative $I_{Kr}$ recordings in the absence or presence of 3 μM LUF7346. No changes were observed on the steady-state activation of $I_{Kr}$ (Fig 1D; $V_{1/2}$: CTR = $-27.52 \pm 1.45$ mV, LUF7346 3 μM = $-26.05 \pm 1.73$ mV, n.s.; $k$: CTR = $5.20 \pm 0.59$, LUF7346 3 μM = $6.15 \pm 0.9$, n.s.; Table 1). By contrast, LUF7346 shifted the steady-state inactivation of $I_{Kr}$ to the right ($V_{1/2}$: CTR = $-38.01 \pm 3.2$ mV, LUF7346 3 μM = $-22.91 \pm 2.0$ mV, $P = 0.0015$), thus extending the range of voltages in which $I_{Kr}$ is available (Fig 1D and Table 2). We further investigated the effect of LUF7346 on $I_{Kr}$ deactivation. Both the fast ($\tau_{fast}$) and the slow ($\tau_{slow}$) components of $I_{Kr}$ deactivation obtained from the fit of the tail current decay were significantly increased so that in the presence of LUF7346, tail currents could not be fitted with a biexponential function at pulse potentials more positive than $-80$ mV (Fig 1E and Table 3).

Taken together, these data indicated that LUF7346 increases $I_{Kr}$ by rightward shifting the voltage dependence of inactivation and slowing deactivation kinetics.

## Electrophysiological characterisation of cardiomyocytes from independent sets of hiPSC isogenic pairs

A summary of the patient hiPSC and hESC lines used in this study is shown in Fig 2A. Three independent, patient-derived and genetically engineered isogenic hiPSCs were included as follows: the

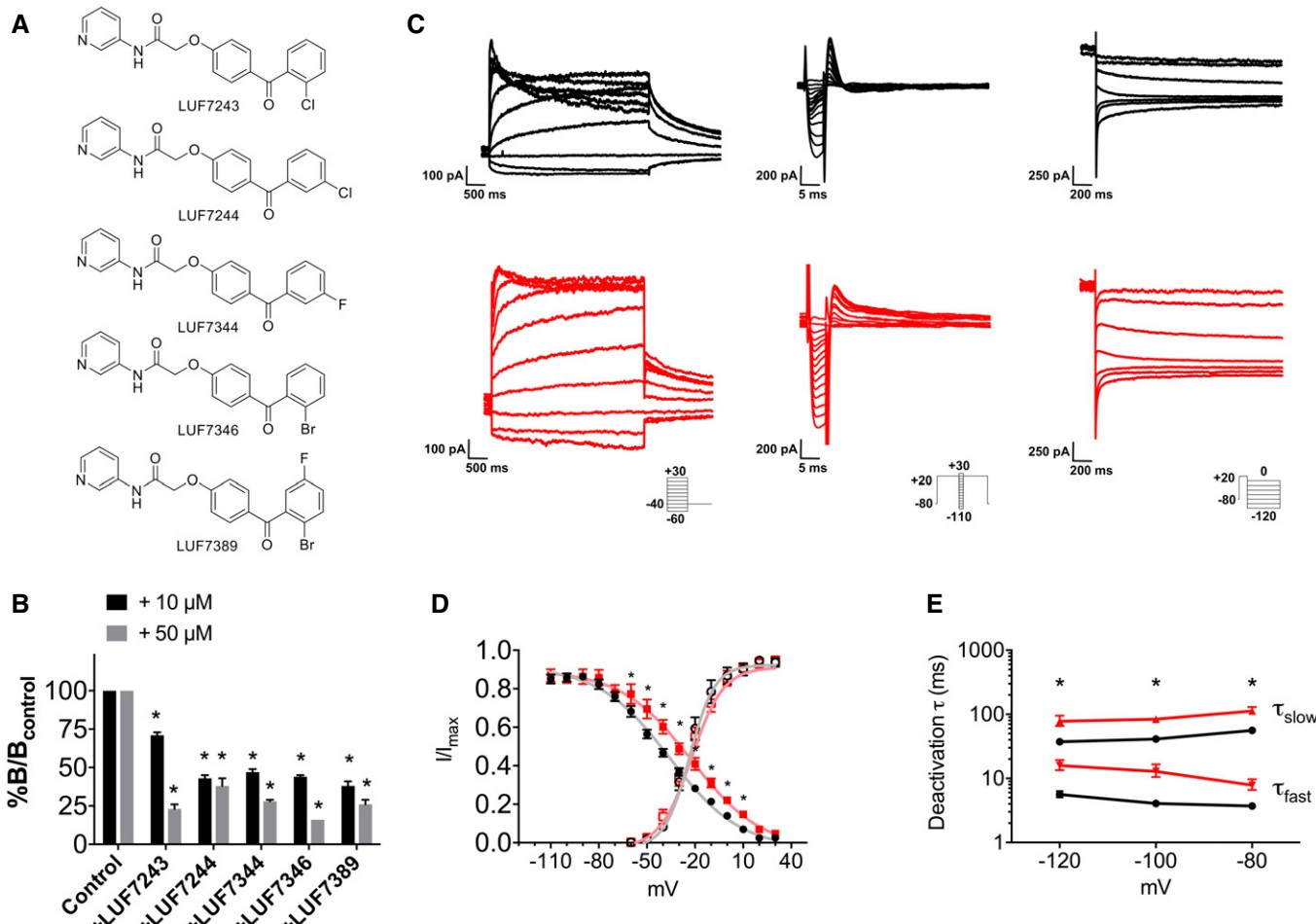

**Figure 1.   The potent hERG allosteric modulator LUF7346 enhances $I_{Kr}$ in HEK293 hERG cells.**

A   Chemical structures of LUF7243, LUF7244, LUF7344, LUF7346 and LUF7389.
B   Percentage of specific binding of [$^3$H]dofetilide to the hERG channel after 6 min of dissociation induced by 10 μM dofetilide in the absence (control) or presence of 10 and 50 μM of LUF compounds. The specific binding of [$^3$H]dofetilide in the absence of test compounds was set as $B_{control}$, while the specific binding in their presence was set as $B$. *$P < 0.05$ versus control; $N = 3$–4.
C   Representative traces of hERG activation (left), inactivation (middle) and deactivation (right) measured in HEK293 hERG cells under baseline conditions (black) and in the presence of 3 μM LUF7346 (red). Insets: voltage-clamp protocols.
D   Steady-state activation (empty symbols) and inactivation (filled symbols) curves for $I_{Kr}$ under baseline conditions (black) and in the presence of 3 μM LUF7346 (red). The corresponding Boltzmann's fittings are superimposed to data points. *$P < 0.05$ versus baseline. $N = 11$–14.
E   Plot of the time constants (τ) of deactivation derived from biexponential fittings under baseline conditions (black) and in the presence of 3 μM LUF7346 (red). *$P < 0.05$ versus respective baseline. $N = 8$.

Data information: (B) Two-tailed unpaired *t*-test. $P < 0.0001$ for all tested LUF compounds compared to the control group at both 10 and 50 μM. (D) Repeated-measures two-way ANOVA with Sidak's multiple comparisons test. Adjusted *P*-values for inactivation: −100 mV: > 0.9999; −90 mV: > 0.9999; −80 mV: 0.3922; −70 mV: 0.9636; −60 mV: 0.0006; −50 mV: < 0.0001; −40 mV: < 0.0001; −30 mV: < 0.0001; −20 mV: < 0.0001; −10 mV: < 0.0001; −10 mV: 0.0011; 0 mV: 0.0111; 10 mV: 0.0343; 20 mV: 0.4871; 30 mV: > 0.9999. Adjusted *P*-values for activation: −60 mV: > 0.9999; −50 mV: 0.9998; −40 mV: > 0.9999; −30 mV: 0.8854; −20 mV: 0.2129; −10 mV: 0.1188; 0 mV: 0.9944; 10 mV: > 0.9999; 20 mV: 0.8340; 30 mV: 0.9841. (E) Paired two-way ANOVA with Sidak's multiple comparisons test. Adjusted *P*-values for deactivation $τ_{fast}$: −120 mV: 0.01, −100 mV: < 0.0001, −80 mV: < 0.0001. Adjusted *P*-values for deactivation $τ_{slow}$: −120 mV: 0.01, −100 mV: < 0.0001, −80 mV: < 0.0001. (B, D, E) Data are expressed and plotted as the mean ± SEM.

LQT1$^{R594Q}$/JLNS$^{R594Q}$ pair [harbouring the heterozygous and homozygous c.1781G > A *KCNQ1* mutation, respectively (Zhang *et al*, 2014)], the LQT2$^{N996I}$/LQT2$^{corr}$ pair [in which the heterozygous *KCNH2* c.2987A > T mutation is maintained or corrected, respectively (Bellin *et al*, 2013)] and a new LQT1$^{R190Q}$/LQT1$^{corr}$ pair (where the heterozygous *KCNQ1* c.569G > A mutation is maintained or corrected, respectively) (Moretti *et al*, 2010; Chen *et al*, 2016). One unrelated wild-type hiPSC line was included as an additional

control (WT, Zhang *et al*, 2014). In addition, the hESC$^{WT}$/hESC-LQT2$^{N996I}$ isogenic pair [in which the heterozygous *KCNH2* c.2987A > T mutation is absent or inserted, respectively (Bellin *et al*, 2013)] was analysed. All of these hPSC lines were differentiated into cardiomyocytes (CMs), which expressed the major ion channels contributing to the AP formation (Fig EV1A) and were electrophysiologically characterised under identical experimental conditions (see Materials and Methods for details). The QT interval

**Table 1.   Steady-state $I_{Kr}$ activation parameters from Boltzmann's fittings in HEK293 hERG cells and in hiPSC-CMs.**

|  | CTR HEK293 hERG | 3 μM LUF7346 HEK293 hERG | CTR hiPSC-CMs | 5 μM LUF7346 hiPSC-CMs |
|---|---|---|---|---|
| $G_{min}$ (nS) | −0.04 ± 0.06 | 0.14 ± 0.1 | 0.11 ± 0.03 | 0.15 ± 0.04 |
| $G_{max}$ (nS) | 4.53 ± 0.52 | 5.57 ± 0.77 | 1.56 ± 0.22 | 2.47 ± 0.32[a] |
| $V_{1/2}$ (mV) | −27.52 ± 1.45 | −26.05 ± 1.73 | −18.40 ± 1.23 | −23.26 ± 0.92[a] |
| $k$ (mV) | 5.20 ± 0.59 | 6.15 ± 0.9 | 5.27 ± 0.56 | 5.18 ± 0.38 |
| $G_{max}$ (nS/pF) | 0.065 ± 0.008 | 0.071 ± 0.01 | 0.02 ± 0.001 | 0.037 ± 0.003[a] |
| $n$ | 11 | 11 | 19 | 19 |

[a]$P < 0.05$ versus CTR hiPSC-CMs.

**Table 2.   Steady-state $I_{Kr}$ inactivation parameters from Boltzmann's fittings in HEK293 hERG cells.**

|  | CTR HEK293 hERG | 3 μM LUF7346 HEK293 hERG |
|---|---|---|
| $V_{1/2}$ (mV) | −38.01 ± 3.22 | −22.91 ± 2.03[a] |
| $dx$ (pA/mV) | 26.19 ± 1.98 | 27.72 ± 3.56 |
| $k$ (mV) | 10.88 ± 1.98 | 10.34 ± 2.89 |
| $n$ | 14 | 14 |

[a]$P < 0.05$ versus CTR HEK293 hERG.

measured with MEA was as expected from the genotype and confirmed previous characterisation (Moretti *et al*, 2010; Bellin *et al*, 2013; Zhang *et al*, 2014), with JLNS[R594Q] > LQT1[R594Q] (275.6 ± 14.7 ms versus 196.4 ± 7.4 ms, $P < 0.0001$), LQT2[N996I] > LQT2[corr] (315.2 ± 25.3 ms versus 227.5 ± 19.2 ms, $P = 0.0151$), hESC[WT] > hESC-LQT2[N996I] (168.1 ± 11.6 ms versus 251.1 ± 21.3 ms, $P = 0.0367$) and LQT1[R190Q] > LQT1[corr] (615.1 ± 103.1 versus 405.8 ± 60.4, $P = 0.0367$; Fig 2B). Of note, the QT interval in control hiPSC-CMs differed from line to line (Fig EV1B, 129.8 ± 8.9 ms WT, 168.1 ± 11.6 ms hESC[WT], 227.5 ± 19.2 ms LQT2[corr], 405.8 ± 60.4 LQT1[corr]). These results suggested a strong influence of the genetic background on the QT interval *in vitro* as such. Furthermore, the RR interval of WT hiPSC-CMs (0.96 ± 0.07 s) was significantly different from those of all the other lines, while the RR was homogeneous within each isogenic pair (LTQ1[R594Q], 2.0 ± 0.13 s, $P < 0.0001$ versus WT; JLNS[R594Q], 2.32 ± 0.21 s, $P = 0.0001$ versus WT; LQT2[corr], 2.41 ± 0.25 s, $P = 0.0006$ versus WT; LQT2[N996I], 2.07 ± 0.22 s, $P = 0.0049$ versus WT; LQT1[corr],

2.79 ± 0.27 s, $P < 0.0001$ versus WT; LQT1[R190Q], 2.1 ± 0.21 s, $P = 0.0198$ versus WT, hESC[WT], 3.7 ± 0.59 s, $P < 0.0001$; hESC[N996I]: 3.1 ± 0.57, $P = 0.0004$; Fig 2B). QT intervals were further corrected with the Bazett's formula (Figs 2B and EV2): 147.4 ± 14.6 ms WT, 148 ± 6.3 ms LQT1[R594Q], 207.8 ± 12.1 ms JLNS[R594Q], 157.6 ± 18.6 ms LQT2[corr], 232.3 ± 19.9 ms LQT2[N996I], 221.3 ± 25.4 ms LQT1[corr], 413 ± 58.4 ms LQT1[R190Q], 97.11 ± 11 ms hESC[WT], 141.9 ± 7.8 ms hESC-LQT2[N996I].

**LUF7346 increases $I_{Kr}$ in patient-specific hiPSC-CMs**

The effects of LUF7346 were further characterised in hiPSC-CMs by voltage clamp on the WT line (Fig 2C–E). LUF7346 at 5 μM significantly increased the $I_{Kr}$ tail current peak (CTR: 1.77 ± 0.18 pA/pF, LUF7346: 3.12 ± 0.27 pA/pF at +10 mV, $P < 0.0001$, Fig 2D) and significantly slowed the fast component of the deactivation τ from −10 mV to +20 mV (τ$_{slow}$ n.s., Fig 2E, Tables 1 and 4). A slight shift in the activation curve towards more negative potentials was also observed (Fig 2D).

Collectively, these data suggested that LUF7346 activates the native hERG channels in hiPSC-CMs. No effects were detected on $I_{Ks}$ and $I_{CaL}$ as measured in the WT-CMs for LUF7346 (Fig EV3).

**LUF7346 rescues the genetic form of LQTS**

Given these strong and specific effects on hERG, we determined whether LUF7346 was able to ameliorate or even rescue the genetic form of LQTS (Fig 3). The pharmacodynamics of LUF7346 were investigated with MEA by applying increasing concentrations of the drug to spontaneously beating hiPSC-CMs and by offline analysis of

**Table 3.   $I_{Kr}$ fast and slow component of deactivation decay from biexponential fittings in HEK293 hERG cells.**

|  | CTR HEK293 hERG | | 3 μM LUF7346 HEK293 hERG | |
|---|---|---|---|---|
| Pulse potential (mV) | τ$_{fast}$ (ms) | τ$_{slow}$ (ms) | τ$_{fast}$ (ms) | τ$_{slow}$ (ms) |
| −120 | 5.60 ± 0.7 | 37.34 ± 3.10 | 10.31 ± 1.74[a] | 77.69 ± 12.08[a] |
| −100 | 4.06 ± 37 | 41.04 ± 1.90 | 6.71 ± 1.40[a] | 83.72 ± 8.80[a] |
| −80 | 3.71 ± 20 | 56.13 ± 4.09 | 4.02 ± 0.67[a] | 112.35 ± 13.61[a] |
| −60 | 65.72 ± 38.57 | 323.63 ± 185.2 | Monoexponential (319.69 ± 29.8) | |
| −40 | 37.26 ± 27.13 | 347.96 ± 187.3 | Monoexponential (388.75 ± 214.9) | |
| −20 | 13.40 ± 9.8 | 248.31 ± 150.7 | Monoexponential (1,316.16 ± 394.9) | |
| $n$ | 8 | 8 | 8 | 8 |

[a]$P < 0.05$ versus the respective value in CTR HEK293 hERG group.

**Figure 2. LUF7346 enhances $I_{Kr}$ in hiPSC-CMs.**

A   Summary of the hPSC lines used in this study. WT, wild-type healthy control; LQT1[R594Q] and JLNS[R594Q], isogenic pair harbouring the heterozygous and the homozygous *KCNQ1* c.1781G > A mutation, respectively; LQT2[N996I] and LQT2[corr], isogenic pair in which the heterozygous c.2987A > T *KCNH2* mutation is present or corrected, respectively; hESC[WT] and hESC-LQT2[N996I], isogenic pair in which *KCNH2* is wild type or the heterozygous c.2987A > T mutation was inserted, respectively; LQT1[R190Q] and LQT1[corr], isogenic pair in which the heterozygous c.569G > A *KCNQ1* mutation is present or corrected, respectively. Arrows indicate the parent line from which each genetically matched isogenic hiPSC were generated.

B   Baseline QT interval (top), baseline RR interval (middle) and baseline QT interval corrected with the Bazett's formula (bottom). Bar graphs are divided by isogenic pairs (LQT1[R594Q] and JLNS[R594Q], left; LQT2[corr], LQT2[N996I], hESC[WT] and hESC-LQT2[N996I], middle; LQT1[corr] and LQT1[R190Q], right), and in each graph, the unrelated WT is shown as a comparison. *N* = 17–53. \*$P$ < 0.05. The colour of the symbol indicates comparisons and relative statistical significance.

C   Representative traces of $I_{Kr}$ steady-state activation in WT hiPSC-CMs in Tyrode (left), in the presence of 5 μM LUF7346 (middle) and after the application of 5 μM E4031 to selectively block $I_{Kr}$ (right). Inset: voltage-clamp protocol.

D   Average I/V relationships (left) and steady-state activation curves with superimposed Boltzmann's fittings (right) under baseline conditions (black) and in the presence of 5 μM LUF7346 (red). \*$P$ < 0.05. *N* = 19.

E   Plot of the time constants (τ) of deactivation derived from biexponential fits (left) and representative examples (right) under baseline conditions (black) and in the presence of 3 μM LUF7346 (red). \*$P$ < 0.05. *N* = 17–21.

Data information: Kruskal–Wallis tests with pairwise Dunn's multiple testing correction. (B) Adjusted $P$-values for QT: WT versus LQT1[R594Q]: 0.0029; WT versus JLNS[R594Q]: < 0.0001; LQT1[R594Q] versus JLNS[R594Q]: < 0.0001. WT versus LQT2[corr]: 0.0238; WT versus LQT2[N996I]: < 0.0001; WT versus hESC-LQT2[N996I]: 0.0034; LQT2[corr] versus LQT[N996I]: 0.0151; LQT2[N996I] versus hESC[WT]: < 0.0001; hESC[WT] versus hESC-LQT2[N996I]: 0.0367. WT versus LQT1[corr]: 0.0098; WT versus LQT1[R190Q]: < 0.0001. LQT1[corr] versus LQT1[R190Q]: 0.0327. Adjusted $P$-values for RR: WT versus LQT1[R594Q]: < 0.0001; WT versus JLNS[R594Q]: < 0.0001. WT versus LQT2[corr]: 0.0001; WT versus LQT2[N996I]: 0.0031; WT versus hESC[WT]: < 0.0001; WT versus hESC-LQT2[N996I]: < 0.0001. WT versus LQT1[corr]: < 0.0001; WT versus LQT1[R190Q]: 0.0012. Adjusted $P$-values for QTc$_B$: WT versus JLNS[R594Q]: 0.0050; LQT1[R594Q] versus JLNS[R594Q]: < 0.0001. WT versus LQT2[N996I]: 0.0074; LQT2[corr] versus LQT2[N996I]: 0.0084. LQT2[N996I] versus hESC[WT] < 0.0001; LQT2[N996I] versus hESC-LQT2[N996I]: 0.0008. WT versus LQT1[corr]: 0.0303; WT versus LQT1[R190Q]: < 0.0001. LQT1[corr] versus LQT1[R190Q]: 0.0018. (D) Paired two-way ANOVA with Sidak's multiple comparisons test. Adjusted $P$-values for I/V plot: −60 mV: 0.9780; −50 mV: > 0.9999; −40 mV: > 0.9999; −30: 0.0037; −20 mV: < 0.0001; −10 mV: < 0.0001; 0 mV: < 0.0001; 10 mV: < 0.0001; 20 mV: < 0.0001; 30 mV: < 0.0001. Adjusted $P$-values for activation: −50 mV: 0.9987; −40 mV: > 0.9999; −30 mV: 0.4519; −20 mV: 0.0013; −10 mV: 0.8323; 0 mV: > 0.9999; 10 mV: 0.9999; 20 mV: 0.5892. (E) Unpaired two-way ANOVA with Sidak's multiple comparisons test. Adjusted $P$-values for deactivation τ$_{fast}$: −10 mV: 0.0469; 0 mV: 0.0358; 10 mV: 0.0103; 20 mV: < 0.0001. Adjusted $P$-values for deactivation τ$_{slow}$: −10 mV: 0.5008; 0 mV: 0.5740; 10 mV: 0.9024; 20 mV: 0.2181. (B, D, E) Data are expressed and plotted as the mean ± SEM.

Source data are available online for this figure.

**Table 4.  I$_{Kr}$ fast ($\tau_{fast}$) and slow ($\tau_{slow}$) components of deactivation decay from biexponential fittings in hiPSC-CMs.**

| Pulse potential (mV) | CTR hiPSC-CMs | | 5 µM LUF7346 hiPSC-CMs | |
|---|---|---|---|---|
| | $\tau_{fast}$ (ms) | $\tau_{slow}$ (ms) | $\tau_{fast}$ (ms) | $\tau_{slow}$ (ms) |
| −10 | 123.39 ± 11.66 | 1,035.90 ± 191.23 | 192.90 ± 19.85[a] | 1,355.18 ± 182.87 |
| 0 | 116.79 ± 11.27 | 911.91 ± 228.17 | 174.04 ± 17.38[a] | 1,528.50 ± 365.89 |
| 10 | 114.54 ± 14.03 | 911.56 ± 157.40 | 179.56 ± 179.56[a] | 1,319.38 ± 159.73 |
| 20 | 116.25 ± 12.44 | 831.62 ± 126.04 | 225.68 ± 39.27[a] | 1,832.61 ± 394.68 |
| n | 21 | 21 | 17 | 17 |

[a]$P < 0.05$ versus the respective value in CTR hiPSC-CMs group.

QT and RR intervals. Figure 3A shows representative FP recordings of LQT2$^{N996I}$-CMs (left) and quantification of dose–response changes in QT interval upon drug application in all hiPSC lines (right). At concentrations of 5 µM and higher, LUF7346 significantly shortened the QT interval in all of the LQTS hiPSC lines and controls analysed, in comparison with their respective baseline; the JLNS line, instead, required a higher drug concentration (10 µM) to undergo a significant QT shortening. At 30 µM, the small molecule LUF7346 induced 50–70% QT shortening. LUF7346 was also proved to be more potent than two known hERG activators, Rottlerin and NS1643 (Fig EV4). As expected by the presence of distinct ion channel mutations and the diverse genetic backgrounds among the hiPSC lines, LUF7346 had heterogeneous effects on RR in the groups analysed (Appendix Fig S2A). In particular, in the LQT1$^{R190Q}$- and LQT1$^{corr}$-CMs (at 20 µM) and in the LQT1$^{R594Q}$- and JLNS$^{R594Q}$-CMs (at 30 µM), we observed a small increase in the RR interval, which occasionally led to cessation of spontaneous beating in hiPSC-CMs.

To evaluate the effect of LUF7346 on isolated hiPSC-CMs in more detail, we measured single-cell APs by patch clamp. The LQT2$^{N996I}$/LQT2$^{corr}$ and LQT1$^{R190Q}$/LQT1$^{corr}$ isogenic pairs were chosen as a representative example of genetic LQTS. LUF7346 significantly shortened the AP duration (APD) at 90% of the repolarisation phase (APD$_{90}$) in LQT2$^{N996I}$-CMs starting from 1 µM (242.2 ± 38.4 ms versus 218.3 ± 37.8 ms); the shortening was more pronounced at 3 µM, where the LQT2$^{N996I}$ APD$_{90}$ became shorter than the control

condition in its isogenic control LQT2$^{corr}$ (242.2 ± 38.4 ms versus 134.9 ± 22.9 ms, Fig 3B and C). Similar effects were observed on AP duration at 70 and 50% of the repolarisation phase (APD$_{70}$ and APD$_{50}$, respectively, Fig 3C). Considering the significantly longer AP recorded in the LQT1$^{R190Q}$/LQT1$^{corr}$ isogenic pair, we decided to apply LUF7346 only at 3 and 5 µM, where the lower concentration already proved efficient in shortening the APD$_{90}$, APD$_{70}$ and APD$_{50}$. No significant influences of LUF7346 were detected on AP amplitude (APA) and diastolic membrane potential (E$_{diast}$) (Fig 3D).

**LUF7346 rescues drug-induced LQTS**

Next, LUF7346 was used for the pharmacological rescue of drug-induced LQTS. For this purpose, the hERG blocker astemizole (AST), an anti-histamine drug withdrawn from the market because of its cardiotoxic effect (Zhou et al, 1999; Chiu et al, 2004), was added to WT hiPSC-CMs (Fig 4). In addition, 100 nM AST was also added to the hiPSC-CMs carrying LQTS mutations, thus simulating in vitro what happens to LQT mutation carriers treated with hERG blocking drugs (Kannankeril et al, 2010; Schwartz, 2015). Figure 4A (left) shows representative MEA recordings in the LQT2$^{N996I}$ line. Firstly, AST treatment induced QT interval prolongation in all of the lines analysed, compared to their respective untreated baseline, as assessed by MEA recordings (Fig 4A, right). Secondly, increasing concentrations of LUF7346 in the presence of 100 nM AST were

**Figure 3.  LUF7346 rescues genetic LQTS.**

A  Representative MEA trace (left) showing the effect of increasing concentrations of LUF7346 on FP contour, measured in LQT2$^{N996I}$-CMs. Average data (right) showing the effect of LUF7346 on QT interval duration relative to baseline in CMs derived from all the hiPSC lines used in this study. *$P < 0.05$ versus baseline. Colour in the heatmap defines the magnitude of QT shortening (blue) after treatment with increasing concentrations of LUF7346. N = 7–14.

B  Representative AP from LQT2$^{corr}$-, LQT2$^{N996I}$-, LQT1$^{corr}$- and LQT1$^{R190Q}$-CMs paced at 1 Hz, under baseline conditions (black) and after application of increasing concentrations of LUF7346 (colour-code is shown).

C  Effect of LUF7346 on APD$_{90}$, APD$_{70}$ and APD$_{50}$, measured in LQT2$^{corr}$- (blue), LQT2$^{N996I}$- (green), LQT1$^{corr}$- (purple) and LQT1$^{R190Q}$-CMs (dark blue) paced at 1 Hz; *$P < 0.05$ versus baseline. The colour of the asterisk indicates comparisons and respective statistical significance. °$P < 0.05$ versus the genetically matched corrected control. #$P < 0.05$ LQT1$^{corr}$ versus LQT2$^{corr}$. N: 12–15.

D  Effect of LUF7346 on APA and E$_{diast}$, measured in LQT2$^{corr}$- (blue), LQT2$^{N996I}$- (green), LQT1$^{corr}$- (purple) and LQT1$^{R190Q}$-CMs (dark blue) paced at 1 Hz. N = 12–15.

Data information: (A) Repeated-measures ANOVA with Sidak's multiple testing correction. Adjusted P-values: LQT1$^{R594Q}$: baseline versus LUF7346 5, 10, 20, 30 µM: < 0.0001. JLNS$^{R594Q}$: baseline versus LUF7346 10 µM: 0.0051. Baseline versus LUF7346 20, 30 µM: < 0.0001. WT: baseline versus LUF7346 5 µM: 0.0329. Baseline versus LUF7346 10, 20, 30 µM: 0.0001. LQT2$^{corr}$: baseline versus LUF7346 10 µM: 0.0205. Baseline versus LUF7346 20, 30 µM: 0.0001. LQT2$^{N996I}$: baseline versus LUF7346 10 µM: 0.0004. Baseline versus LUF7346 20, 30 µM: < 0.0001. LQT1$^{R190Q}$: baseline versus LUF7346 5 µM: 0.0042. Baseline versus LUF7346 10, 20, 30 µM: 0.0001. LQT1$^{corr}$: baseline versus LUF7346 10 µM: 0.0006. Baseline versus LUF7346 20, 30 µM: < 0.0001. (C) Paired two-way ANOVA with Holm–Sidak's multiple comparisons test. APD$_{90}$: Tyr LQT2$^{corr}$ versus Tyr LQT2$^{N996I}$: 0.0013. Tyr LQT1$^{corr}$ versus Tyr LQT1$^{R190Q}$: < 0.0001. Tyr LQT2$^{corr}$ versus Tyr LQT1$^{R190Q}$: < 0.0001. Tyr LQT2$^{corr}$ versus Tyr LQT1$^{corr}$: 0.0278. Tyr LQT2$^{N996I}$ versus Tyr LQT1$^{R190Q}$: 0.0008. Tyr LQT2$^{N996I}$ versus LQT1$^{corr}$: 0.31071. Tyr LQT1$^{R190Q}$ versus Tyr LQT1$^{corr}$: < 0.0001. LQT2$^{corr}$: Tyr versus LUF7346 3 µM: 0.0433. Tyr versus LUF7346 5 µM: < 0.0001. LQT2$^{N996I}$: Tyr versus LUF7346 3 µM, 5 µM: < 0.0001. APD$_{70}$: Tyr LQT2$^{corr}$ versus Tyr LQT2$^{N996I}$: 0.0478. Tyr versus LUF7346 5 µM: 0.0018. LQT2$^{N996I}$: Tyr versus LUF7346 3 µM, 5 µM: < 0.0001. APD$_{50}$: Tyr versus LUF7346 5 µM: 0.0023. LQT2$^{N996I}$: Tyr versus LUF7346 3 µM, 5 µM: < 0.0001. LQT1$^{corr}$: Tyr versus LUF7346 3 µM: 0.0006; Tyr versus LUF7346 5 µM: < 0.0001. LQT1$^{R190Q}$: Tyr versus LUF7346 3 µM, 5 µM: < 0.0001. (D) Repeated-measures two-way ANOVA with Holm–Sidak's multiple testing correction. All comparisons were not significant. (C, D) Data are expressed and plotted as the mean ± SEM.

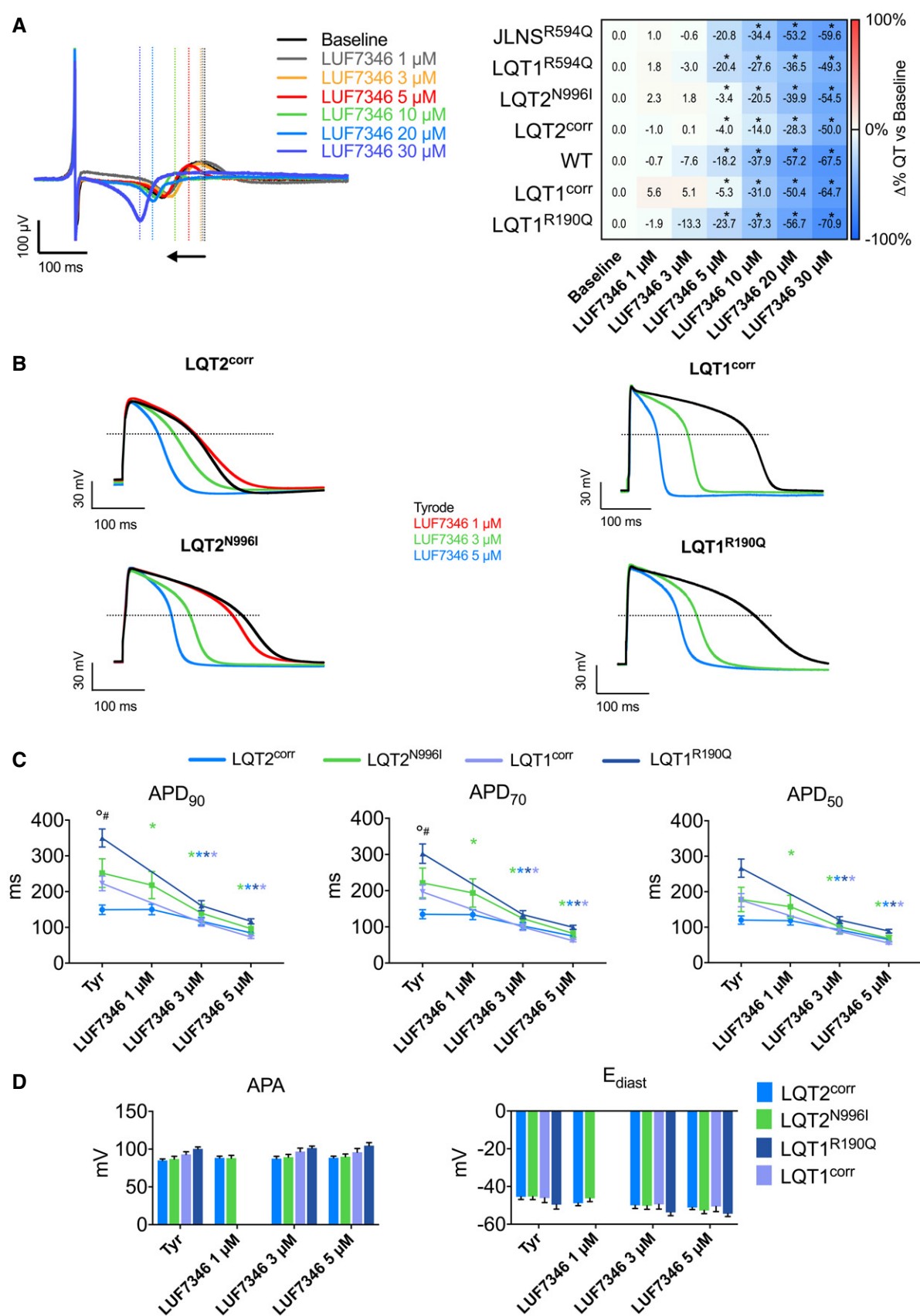

**Figure 3.**

applied. LUF7346 at 10 μM was sufficient to reverse the QT interval prolongation induced by AST; however, this concentration was not able to normalise the QT interval further to baseline values. At 20 μM, instead, LUF7346 significantly shortened the QT interval compared to both AST and baseline in all the lines, with the exception of the JLNS hiPSC-CMs that carry a homozygous *KCNQ1* mutation. By increasing the dose to 30 μM, significant shortening of the QT interval was observed in all the lines tested, including the JLNS hiPSC-CMs ($-20\%/-50\%$ compared to baseline, $-60\%/-70\%$ compared to AST). AST did not have any effect on the RR interval in any of the tested lines (Appendix Fig S2B). LUF7346 instead displayed a small, although not statistically significant, RR prolongation in almost all of the lines, accompanied by cessation of beating in some clusters but only at the highest concentrations.

To analyse the effects of AST and LUF7346 on single cells further, we measured AP with patch clamp in isolated hiPSC-CMs from the LQT1$^{R594Q}$/JLNS$^{R594Q}$ isogenic pair, as an illustrative example for the combination of genetic and drug-induced LQTS, and from the WT line, representing drug-induced LQTS (Fig 4B). AST 100 nM prolonged the APD$_{90}$ in WT, LQT1$^{R594Q}$ and JLNS$^{R594Q}$ hiPSC-CMs by $59.25 \pm 8.8\%$, $45.25 \pm 4.5\%$ and $46.08 \pm 3.8\%$, respectively. Application of 1 μM LUF7346 in the presence of 100 nM AST failed to restore baseline values. About 3 μM LUF7346, instead, massively shortened APD$_{90}$ values in WT hiPSC-CMs, completely restored them in LQT1$^{R594Q}$-CMs but failed to significantly shorten AP in JLNS$^{R594Q}$-CMs. As expected, the severe QT prolongation in JLNS$^{R594Q}$ hiPSC-CMs required a slightly higher concentration of LUF7346 (5 μM) to be rescued. Finally, LUF7346 at 10 μM induced massive APD$_{90}$ shortening in both cell lines. APD$_{70}$ and APD$_{50}$ were also significantly shortened starting from a concentration of 3 μM in both cell lines. In agreement with the results for LUF7346 alone, no effects were

recorded on APA or E$_{diast}$. Interestingly, AST also induced arrhythmic events in JLNS$^{R594Q}$-CMs (in 20.6% of the cells) but not in LQT1$^{R594Q}$-CMs, as evidenced by the presence of early after depolarisations (EADs, Fig 5A); this reflects the increased susceptibility of JLNS patients to hERG blockers (Priori *et al*, 1999). Notably, LUF7346 was able to completely abolish the AST-induced EADs recorded in the JLNS$^{R594Q}$ hiPSC-CMs in a concentration-dependent manner. Increased proarrhythmic risk has often been associated with increased beat-to-beat variability of repolarisation duration (BVR) (Thomsen *et al*, 2006; Altomare *et al*, 2015). Therefore, we analysed the BVR of QT intervals using MEA and quantified it as short-term variability (STV, see Materials and Methods) in all isogenic hiPSC-CMs pairs. Since RR intervals were not different within each isogenic pair, this allowed a good approximation of BVR measured *in vivo* (Thomsen *et al*, 2006). This analysis proved reliable for revealing the severity of the genotype within each isogenic pair, since higher STV values were recorded in the mutated lines (Fig 5B). AST significantly increased STV compared to baseline values in JLNS$^{R594Q}$- ($+28.65 \pm 9.54\%$), LQT2$^{N996I}$- ($+52.18 \pm 20.89\%$) and LQT1$^{R190Q}$-CMs ($+53.62 \pm 14.72\%$) (Fig 5C). No significant effects were recorded in LQT1$^{R594Q}$- ($+14.64 \pm 14.77\%$), LQT2$^{corr}$- ($9.02 \pm 13.6\%$), LQT1$^{corr}$ ($-8.288 \pm 14.83\%$) and WT-CMs (Figs 5C and EV5). The addition of 20 μM LUF7346 in the presence of AST resulted in a significant decrease of STV (from $-19$ to $-50\%$ versus baseline) in all of the lines except LQT1$^{corr}$, where changes did not reach statistical significance.

## Discussion

Differentiated derivatives of disease-specific hiPSCs have already demonstrated their usefulness for *in vitro* drug repurposing

---

**Figure 4. LUF7346 rescues drug-induced LQTS in both wild-type and LQTS genetic backgrounds.**

A  Representative MEA trace (left) showing the effect of 100 nM AST and increasing concentrations of LUF7346 in the presence of AST on FP contour, measured in LQT1$^{R594Q}$-CMs. Average data (right) of the effect of AST and LUF7346 in the presence of AST on QT interval duration compared to baseline in CMs derived from all the hiPSC lines used in this study. *$P < 0.05$ versus respective baseline. °$P < 0.05$ versus AST. Colour in the heatmap defines the magnitude of QT prolongation (red) and QT shortening (blue), respectively, after treatment with AST and with increasing concentrations of LUF7346 in the presence of AST. N = 5–10.

B  Representative AP from WT-, LQT1$^{R594Q}$- and JLNS$^{R594Q}$-CMs, paced at 1 Hz, under baseline conditions (black), after application of 100 nM AST (red) and after addition of increasing concentrations of LUF7346 in the presence of AST (colour-code is shown).

C  Effect of AST and LUF7346 in the presence of AST on APD$_{90}$, APD$_{70}$ and APD$_{50}$ in WT- (grey), LQT1$^{R594Q}$- (red) and JLNS$^{R594Q}$-CMs (black) paced at 1 Hz. *$P < 0.05$ versus respective baseline; °$P < 0.05$ versus AST. N = 11–16.

D  Effect of AST and LUF7346 in the presence of AST on APA and E$_{diast}$ in WT- (grey), LQT1$^{R594Q}$- (red) and JLNS$^{R594Q}$-CMs (black) paced at 1 Hz. N: 11–16.

Data information: (A) Repeated-measures one-way ANOVA with Holm–Sidak's multiple testing correction: LQT1$^{R594Q}$: baseline versus AST: 0.0038, baseline versus AST+LUF7346 20 μM: 0.0007. Baseline versus AST+LUF7346 30 μM: 0.0022. AST versus AST+LUF7346 20 μM: < 0.0001; AST versus AST+LUF7346 30 μM: < 0.0001. JLNSR$^{594Q}$: baseline versus AST: 0.0020; baseline versus AST+LUF7346 20 μM: 0.0195; baseline versus AST+LUF7346 30 μM: 0.0009; AST versus AST+LUF7346 20 μM: 0.0002; AST versus AST+LUF7346 30 μM: < 0.0001. WT: baseline versus AST: 0.0388; baseline versus AST+LUF7346 20 μM: 0.0430; baseline versus AST+LUF7346 30 μM: 0.0021; AST versus AST+LUF7346 20 μM: 0.0282; AST versus AST+LUF7346 30 μM: 0.0015. LQT2$^{corr}$: baseline versus AST: 0.0202; baseline versus AST+LUF7346 30 μM: < 0.0001; AST versus AST+LUF7346 20 μM: 0.0203; AST versus AST+LUF7346 30 μM: 0.0013. LQT2$^{N996I}$: baseline versus AST: 0.0178; baseline versus AST+LUF7346 20 μM: 0.0058; baseline versus AST+LUF7346 30 μM: < 0.0001; AST versus AST+LUF7346 20 μM: 0.0001; AST versus AST+LUF7346 30 μM: < 0.0001. LQT1$^{R190Q}$: baseline versus AST: 0.0466; baseline versus AST+LUF7346 20 μM: 0.0268; baseline versus AST+LUF7346 30 μM: 0.0001; AST versus AST+LUF7346 20 μM: 0.0160; AST versus AST+LUF7346 30 μM: 0.0138. LQT1$^{corr}$: baseline versus AST: 0.0374; baseline versus AST+LUF7346 20 μM: 0.0493; baseline versus AST+LUF7346 30 μM: < 0.0001; AST versus AST+LUF7346 20 μM: 0.0012; AST versus AST+LUF7346 30 μM: < 0.0001. (C) Adjusted *P*-values, respectively, for JLNS$^{R594Q}$; LQT1$^{R594Q}$; WT: APD$_{90}$: Tyr versus AST: 0.0017; 0.0026; 0.0012. Tyr versus AST+LUF7346 1 μM: 0.0448; 0.0027; 0.0002. Tyr versus AST+LUF7346 3 μM: 0.0017; 0.3097; 0.1147. Tyr versus AST+LUF7346 5 μM: 0.9974; 0.2752; 0.0404. Tyr versus AST+LUF7346 10 μM: 0.0145; 0.0003; n/a. AST versus AST+LUF7346 3 μM: 0.0215; < 0.0001; 0.0289. AST versus AST+LUF7346 5 μM: 0.0144; 0.0007; 0.0029. AST versus AST+LUF7346 10 μM: 0.0045; < 0.0001; n/a. APD$_{70}$: Tyr versus AST: 0.0078; 0.0041; 0.0294. Tyr versus AST+LUF7346 1 μM: 0.0488; 0.0266; 0.0452. Tyr versus AST+LUF7346 3 μM: 0.9527; 0.2919;0.4299. Tyr versus AST+LUF7346 5 μM: 0.3003; 0.0039; < 0.0001. Tyr versus AST+LUF7346 10 μM: <0.0001; 0.0009; n/a. AST versus AST+LUF7346 3 μM: 0.0267; < 0.0001; 0.0238. AST versus AST+LUF7346 5 μM: 0.0032; 0.0022; 0.0048. AST versus AST+LUF7346 10 μM: 0.0002; 0.0008; n/a. APD$_{50}$: Tyr versus AST: 0.0385; 0.0472; 0.0055. Tyr versus AST+LUF7346 1 μM: 0.0429; 0.2383; 0.0210. Tyr versus AST+LUF7346 3 μM: 0.8840; 0.0650; 0.2113. Tyr versus AST+LUF7346 5 μM: 0.0470; 0.0023; 0.0020. Tyr versus AST+LUF7346 10 μM: 0.0012; 0.0002; n/a. AST versus AST+LUF7346 3 μM: 0.0191; < 0.0001; 0.0027. AST versus AST+LUF7346 5 μM: 0.0241; 0.0014; 0.0006. AST versus AST+LUF7346 10 μM: 0.0028; 0.0002; n/a. Comparisons not indicated were not statistically significant. (D) One-way ANOVA with Holm–Sidak's multiple testing correction. All comparisons were not significant. (C, D) Data are expressed and plotted as the mean ± SEM.

     

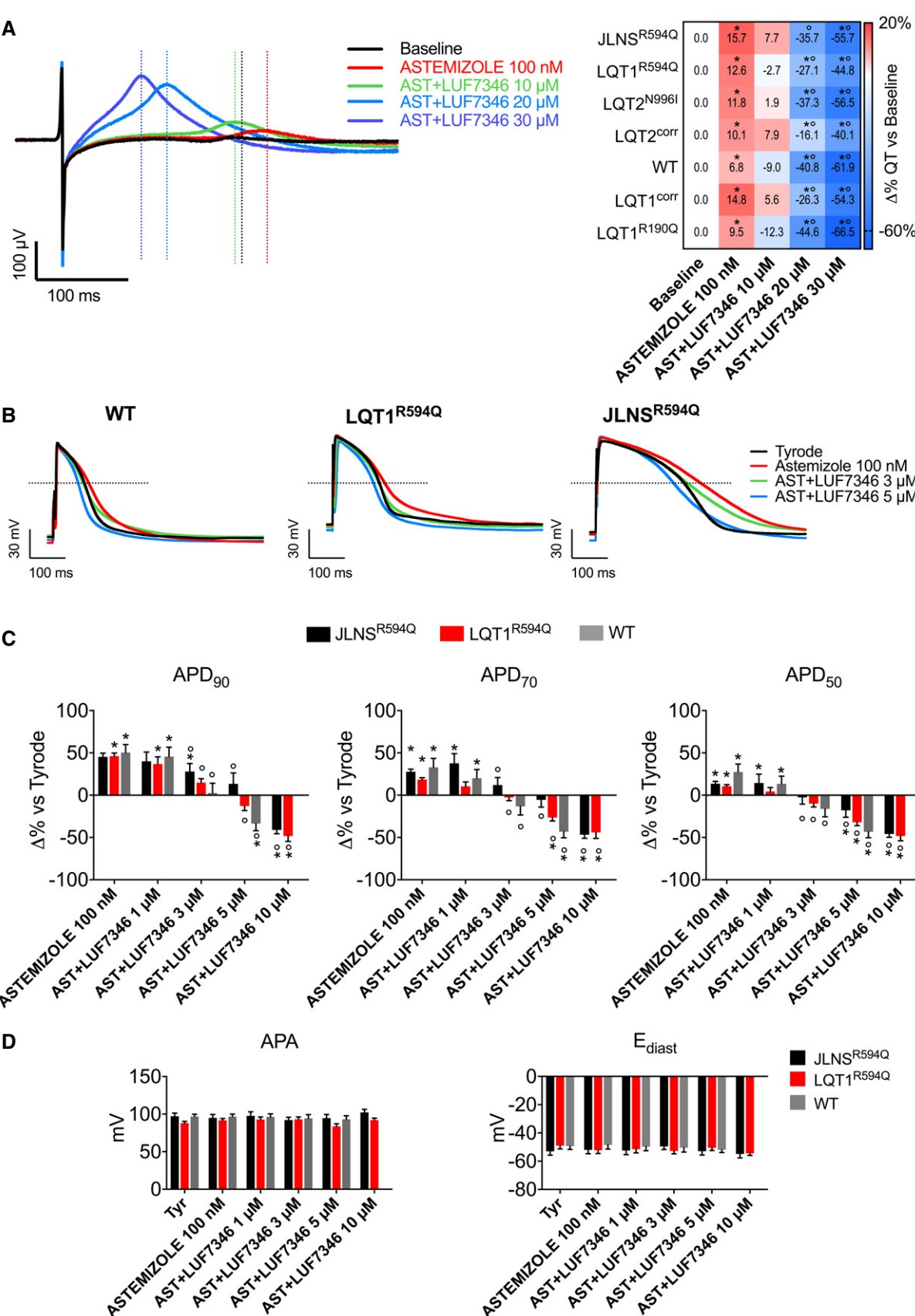

**Figure 4.**

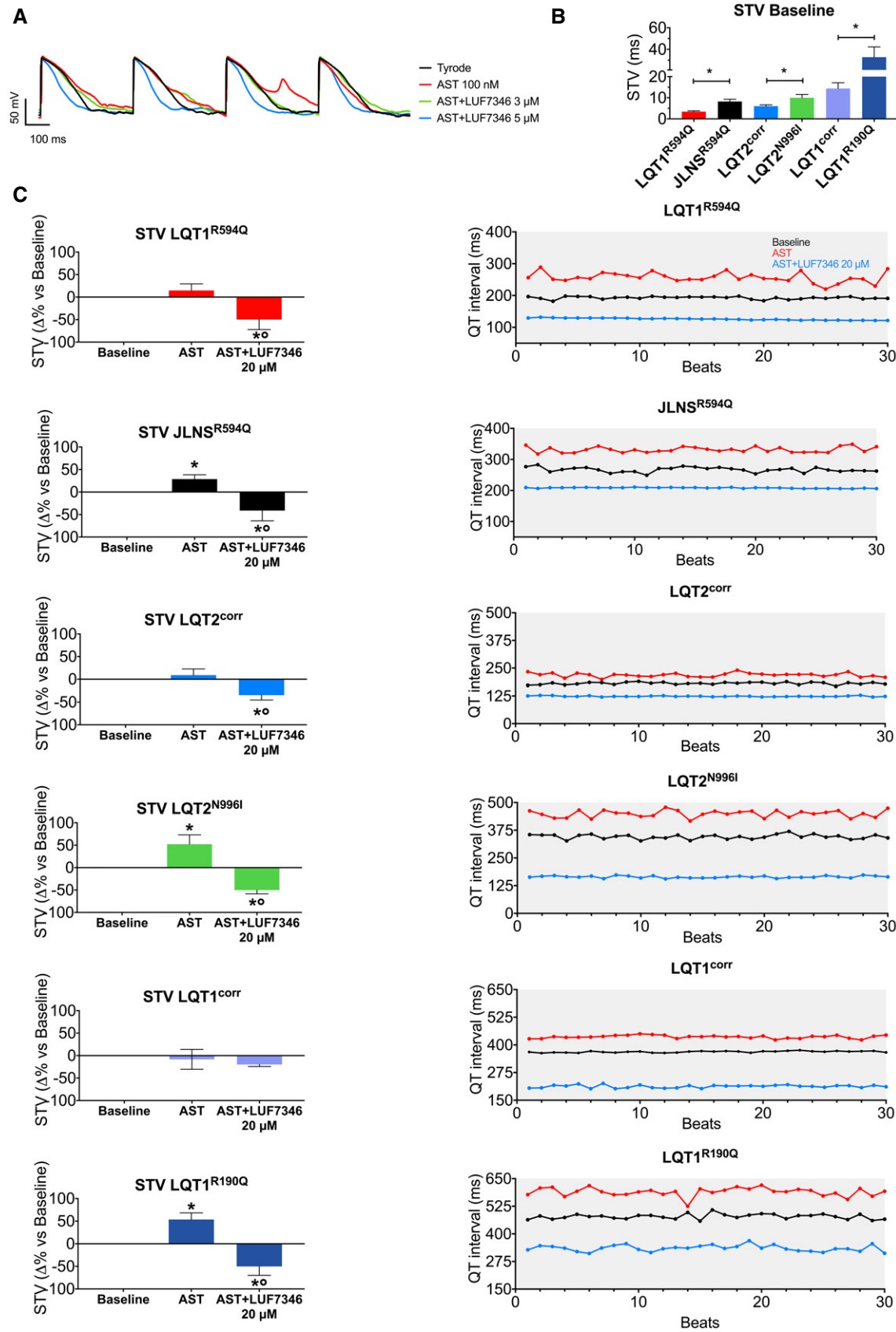

**Figure 5.**

◀

**Figure 5.  LUF7346 suppresses arrhythmogenic events in drug-induced LQTS.**

A   Representative AP traces measured in JLNS$^{R594Q}$-CMs paced at 1 Hz, under baseline conditions (black) and after application of either AST alone (red), AST+LUF7346 3 µM (green) and AST+LUF7346 5 µM (blue).

B   STV values calculated on spontaneously beating hiPSC-CM clusters over 30 consecutive beats as measured with MEA. $N$ = 7–24. *$P$ < 0.05 versus respective isogenic pair.

C   STV changes (left) and representative examples of QT interval oscillations plotted over 30 consecutive beats (right) in the presence of AST and AST+LUF7346 20 µM. *$P$ < 0.05 versus baseline. °$P$ < 0.05 versus AST. $N$ = 7–24.

Data information: (B) Unpaired $t$-tests within each isogenic pair: LQT1$^{R594Q}$-JLNS$^{R594Q}$: $P$ < 0.0001; LQT2$^{corr}$-LQT2$^{N996I}$: 0.0354; LQT1$^{corr}$-LQT1R$^{190Q}$: 0.0364. (C): unpaired one-way ANOVA with Holm–Sidak's multiple comparisons test. LQT1$^{R594Q}$: baseline versus AST: 0.254, baseline versus LUF: 0.0305, AST versus LUF: 0.0087. JLNS$^{R594Q}$: baseline versus AST: 0.0288, baseline versus LUF: 0.0288, AST versus LUF: 0.0003. LQT2$^{N996I}$: baseline versus AST: 0.0192, baseline versus LUF: 0.0192, AST versus LUF: < 0.0001. LQT2$^{corr}$: baseline versus AST: 0.5314, baseline versus LUF: 0.0493, AST versus LUF: 0.0191. LQT1$^{corr}$: baseline versus AST: 0.6821, baseline versus LUF: 0.4582, AST versus LUF: 0.6821. LQT1$^{R190Q}$: baseline versus AST: 0.0161, baseline versus LUF: 0.0161, AST versus LUF: 0.0004. (B, C) Data are expressed and plotted as the mean ± SEM.

(Wainger *et al*, 2014; McNeish *et al*, 2015) and LQTS hiPSC-CMs has proven to be a valuable tool in studying the effects of small molecules (Bellin *et al*, 2012; Zhang *et al*, 2012; Navarrete *et al*, 2013; Sinnecker *et al*, 2014; Sallam *et al*, 2015). However, only rarely have hiPSC-CMs so far been used for thoroughly evaluating new chemical entities and even less frequently have genetically matched controls been used for drug activity evaluation (Wang *et al*, 2014b). Here, we demonstrated the application of a panel of isogenic hPSC-CMs in a drug discovery pipeline for the rescue of both genetic and acquired LQTS. Increased arrhythmia susceptibility to hERG blockers in patients with LQTS genetic backgrounds was also recapitulated in their hiPSC-CMs and rescued by the small molecule LUF7346. This study provides novel perspectives for the treatment of LQTS and strong evidence that hPSC technology can be used in drug discovery.

Several strategies have been used in attempts to correct the LQTS phenotype in hiPSC-CMs; these include treatment with direct/indirect blockers of depolarising currents [e.g. nifedipine (Spencer *et al*, 2014), propranolol (Mehta *et al*, 2013)], application of activators of repolarising currents (Casis *et al*, 2006; Zhang *et al*, 2012; Matsa *et al*, 2014; Wang *et al*, 2014b) and allele-specific RNA interference (Matsa *et al*, 2014). In the present study, we tested a novel molecule (LUF7346) able to modulate hERG channel activity allosterically. LUF7346 acts as a type-1 hERG activator (Perry *et al*, 2009, 2010), by positively shifting the steady-state inactivation curve (thus increasing $I_{Kr}$ window current) and by slowing $I_{Kr}$ deactivation in a voltage-dependent manner (enhancing the amount of current available at a certain membrane potential). Most importantly, LUF7346 appeared to be very specific for $I_{Kr}$, since it did not affect $I_{Ks}$, $I_{CaL}$ or $dV/dt_{max}$ at the working concentrations tested, and it seemed to have negligible effects on $I_{Na}$ and $I_{K1}$, since it did not alter the APA and $E_{diast}$ (Figs 3 and EV3). These results are encouraging for its possible translational application, although they need to be further validated in clinical studies.

Genetic background can play an essential role in shaping disease traits, particularly in monogenic disorders. By using identical experimental conditions for all the hPSC lines analysed, the individual genetic background was excluded as variable in the pairwise comparisons. Interestingly, significant differences in basal QT intervals and RR (and QTc) were observed among the four *bona fide* wild-type lines, confirming that the use of a random selection of wild-type genetic backgrounds is insufficient to justify the term "healthy control". In addition, when LUF7346 was added to rescue the genetic form of LQTS, some differences among control lines were observed in their response to the compound (Appendix

Fig S3), supporting the idea that several different controls might be more informative in the context of drug testing. One limitation that our study shares with the vast majority of the literature is that all analyses were done using one clone per group. We anticipate that this will possibly change as automatic phenotyping and genetic manipulation techniques further improve in terms of cost, time and efficiency. As a cautionary note for any study quantifying QT interval in hPSC-CMs, is our observation that common QT interval correction methods with Bazett's or Fridericia's formulae, routinely used in clinics and developed for rate responses within the clinical range, might be misleading when applied *in vitro*; major-axis regression analysis indicated that hiPSC-CMs derived from distinct lines behave differently in response to the beating frequency (Fig EV2 and Appendix Table S2) suggesting that the classical QT correction methods may not be broadly applicable *in vitro* to all hiPSC lines and should therefore be used carefully. A detailed analysis of single-cell gene and protein expression along with electrophysiological characterisation of hiPSC-CMs may shed light on the differences in QT observed among different lines in future studies. Some variability among the isogenic pairs tested was also observed in the QT prolongation after hERG block (Appendix Fig S4) confirming the important role of genetic background in drug responses. Asymptomatic or borderline LQTS mutation carriers are highly represented in the LQTS population [36 and 19%, respectively, for LQT1 and LQT2 (Schwartz *et al*, 2012)], and they are particularly susceptible to hERG blockers (Fitzgerald & Ackerman, 2005). In fact, it is suspected that drug-induced LQTS often results from unmasking an asymptomatic congenital predisposition (Saenen & Vrints, 2008; Schwartz, 2015). By combining both congenital and drug-induced LQTS as exemplified here with AST treatment, we illustrated the value of hiPSC-CMs as tools for developing precision medicine (Ashley, 2015), introducing a new aspect of the clinical complexity into the link between bench and bedside.

The generation of isogenic controls by gene repair has already proven successful in identifying causative mutations, especially in the case of mild phenotypes (Bellin *et al*, 2013; Wang *et al*, 2014a). Similarly, genetically matched lines act as internal references during drug testing (Wainger *et al*, 2014) and may be essential to determine the active concentration range. Here, as a proof of principle, isogenically matched lines have been used as an internal reference for QT/APD normalisation in the settings of genetic and drug-induced LQTS. As shown in Fig 3B and C, the selection of a genetically matched control is essential for both disease modelling (i.e. no differences in basal APD would be detected between LQT2$^{N996I}$ and LQT1$^{corr}$) and drug screening (if LQT2$^{corr}$ were chosen as control for

the LQT1$^{R190Q}$-CMs, 3 μM of LUF7346 would seem necessary to normalise the AP, while this drug concentration is shortening the AP beyond the value of the isogenically matched LQT1$^{corr}$-CMs). The normalisation effect of LUF7346 within each isogenic hiPSC-CM pair revealed that concentrations between 1 and 3 μM were sufficient to induce significant APD shortening. Importantly, the active concentration of LUF7346 that we identified is from 5 to 15 times lower than previously reported hERG activators in human (Kang et al, 2005; Zhang et al, 2014) or rodent (Yu et al, 2016) cells, although direct comparisons should be made under identical experimental conditions. In addition, LUF7346 alone also affected AP duration, in contrast to other hERG activators (Kang et al, 2005). As for previous observations with hERG blockers (Bellin et al, 2013), the effects of the hERG allosteric modulator LUF7346 were larger in LQT2$^{N996I}$ and LQT1$^{R190Q}$ compared to LQT2$^{corr}$ and LQT1$^{corr}$, respectively, while at higher concentrations, the APD end point converged. In conditions of unstable repolarisation phase, as measured in the mutated CMs (Fig 5B and C), even small currents can induce substantial changes in APD (Virág et al, 2009). Furthermore, I$_{Ks}$ might be more active in LQT2$^{N996I}$ under basal conditions as a compensatory mechanism subsequent to deregulated repolarisation reserve (Braam et al, 2013); in these settings, I$_{Kr}$ activation induced by LUF7346 could cause greater AP shortening.

Despite their apparent value demonstrated here, hPSC-CMs are still limited for some applications by their immature electrophysiological phenotype (Veerman et al, 2015). All the lines analysed by patch clamp tended to be hyperpolarised by LUF7346, in accordance with the strong reliance of E$_{diast}$ in hiPSC-CMs on I$_{Kr}$ (Doss et al, 2012) with slight enhancement promoting hyperpolarisation (Fig EV3D). The effect of LUF7346 on I$_{Kr}$ deactivation was clearly visible as a sustained diastolic outward current, which explains also the apparent, although not statistically significant, increases in dV/dt$_{max}$ and why LUF7346 at high concentrations seldom inhibited spontaneous beating of hiPSC-CMs. Excessive increase in I$_{Kr}$ by LUF7346 may have polarised the membrane and made the clusters quiescent. The technical difficulties of stimulating clusters on the MEA did not allow confirmation of this hypothesis; however, this phenomenon was not observed in isolated hiPSC-CMs paced at 1 Hz.

When drug-induced LQTS was simulated on MEA, the I$_{Kr}$ blockade induced by AST was completely reverted by LUF7346 treatment, confirming the negative allosteric effects of LUF7346 on the binding of prototypical hERG blockers at the channel (Yu et al, 2016). The binding of LUF7346 to the channel at an allosteric site (i.e. a site topologically distinct from where classic hERG blockers bind) triggers a conformational change within the channel, ultimately causing an acceleration of the AST dissociation rate and thus a lower activity at the channel (Christopoulos et al, 2014). In contrast with findings described by others (Matsa et al, 2011), hERG modulation by LUF7346 completely reversed the severe effects of the potassium channel blocker AST in LQT2$^{N996I}$ hiPSC-CMs, confirming the efficacy of LUF7346 compared to other molecules reported so far in the literature. A different degree of sensitivity/severity between JLNS$^{R594Q}$ and LQT1$^{R594Q}$ was also reflected in a more potent effect of LUF7346 on the LQT1$^{R594Q}$ hiPSC-CMs.

One of the risks of treating LQTS with hERG activators is excessive shortening of the QT interval. Here, we observed steep dose–response curves for LUF7346, which could make it difficult to find the right dose for in vitro testing that does not generate short QT

(SQT) (Sanguinetti, 2014). This should be taken into account when translating these results to in vivo experiments or clinical use, even though hERG activator-induced SQT has proven to have milder consequences than previously hypothesised (Grunnet et al, 2008; van der Linde et al, 2008). However, prolonged RR interval and cessation of spontaneous beating as detected in our assays might result in more severe in vivo effects, thus forming an obstacle for clinical translation. Nevertheless, the chronic effects of LUF7346 at 24 and 48 h confirmed its high efficacy and the stability of its effect at physiological temperature (Fig EV6).

In conclusion, libraries of hiPSC-CMs are beginning to become excellent resources as human models for predicting drug responses and cardiotoxicity, as recently demonstrated by the correlation between cardiac doxorubicin sensitivity in patients and their derivative hiPSC-CMs (Burridge et al, 2016). In combination with precision medicine, it is highly likely that prediction of cardiotoxicity from hERG block will be feasible and further exploration encouraged for future regulatory implementation. The use of hiPSC pairs as exemplified here, in combination with in vitro hERG assays, in silico data and large animal studies, as proposed for the comprehensive in vitro proarrhythmic sssay (CiPA, Sager et al, 2014; Fermini et al, 2015), will likely contribute to the development of precision medicine and, eventually, to the reduction of attrition in drug discovery.

## Materials and Methods

### Drugs

The synthesis and chemical analysis of all LUF compounds were detailed in our previously published paper (Yu et al, 2015). Astemizole was purchased from Sigma-Aldrich (MO, USA). Tritium-labelled dofetilide (specific activity: 82.3 Ci/mmol) was obtained from PerkinElmer (MA, USA). All these drugs were dissolved in DMSO to obtain 10 and 100 mM stock solutions. NS1643 (Santa Cruz Biotech, USA) was dissolved in pure EtOH according to the manufacturer's datasheet to obtain a 100 mM stock solution. Rottlerin (Santa Cruz Biotech, USA) was dissolved in DMSO according to the manufacturer's datasheet to obtain a 50 mM stock solution.

All the other chemicals were of analytical grade and obtained from standard commercial sources.

### Cell culture

For in vitro binding assays, HEK293 cells stably expressing hERG 1 channel (HEK293 hERG) were cultured as previously described (Yu et al, 2014). For electrophysiology, HEK293 hERG were cultured in DMEM with 20% FCS, 1:500 penicillin/streptomycin, 1:100 L-glutamine and passaged twice a week.

hPSCs were cultured and differentiated into cardiomyocytes as previously described (Takahashi et al, 2007; Dambrot et al, 2014). Briefly, hPSCs were maintained on MEF feeders in DMEM/F12 medium supplemented with 20% KSR, 2 mM L-glutamine, 0.1 mM non-essential amino acids, 50 U/ml penicillin, 50 μg/ml streptomycin and 10 ng/ml human b-FGF (PeproTech, UK). For cardiac induction, hiPSCs (~3 × 10$^4$ cells/cm$^2$) were plated on Matrigel-coated (BD Biosciences, NJ, USA) plates 1 day before starting the differentiation. One day after seeding, medium was changed to

low-insulin BPEL medium (Ng *et al*, 2008) with the addition of the following factors: day 0 to day 3, 20 ng/ml BMP4 (R&D Systems, MN, USA), 20 ng/ml activin A (Miltenyi Biotec, D) and 1.5 μM GSK3 inhibitor CHIR99021 (Axon Medchem, DE); day 4 to day 6, 5 μM XAV939 Wnt inhibitor (Tocris Bioscience, UK). Medium was further changed every 3–4 days and beating hiPSC-CMs appeared from day 10 onwards. CMs 20–30 days old were dissociated for MEA analysis and single-cell electrophysiology using Tryple Select 1X (Gibco Life Technologies, MA, USA). Correction of the c.569G > A *KCNQ1* mutation was performed by conventional homologous recombination as previously described (Bellin *et al*, 2013).

### *In vitro* binding assay

Membranes from HEK293 cells overexpressing hERG were prepared and stored as described previously (Yu *et al*, 2014). Single-point dissociation assay was performed in incubation buffer (10 mM HEPES, 130 mM NaCl, 60 mM KCl, 0.8 mM MgCl$_2$, 1 mM EGTA, 10 mM glucose, 0.1% BSA, pH 7.4) as detailed previously (Yu *et al*, 2015). Briefly, the dissociation of [$^3$H]dofetilide was initiated by addition of 10 μM dofetilide in the absence (control) or presence of 10 or 50 μM LUF compounds after pre-incubation at 25°C for 2 h. After 6 min of dissociation, incubations were terminated by dilution with ice-cold wash buffer (25 mM Tris–HCl, 130 mM NaCl, 60 mM KCl, 0.8 mM MgCl$_2$, 0.05 mM CaCl$_2$, 0.05% BSA, pH 7.4). Separation of bound from free radioligand was performed by rapid filtration through a 96-well GF/B filter plate using a PerkinElmer Filtermate-harvester PerkinElmer (MA, USA). The filter-bound radioactivity was determined by scintillation spectrometry using the P-E 1450 Microbeta Wallac Trilux scintillation counter (PerkinElmer, MA, USA) after addition of 25 μl Microscint and 2-h extraction. Data were analysed with GraphPad Prism 6.0 or 7.0 (GraphPad Software, CA, USA) and R 3.2.4 (The R Foundation for Statistical Computing).

### Multielectrode arrays

Multielectrode array (MEA) experiments were performed using a 64 electrodes USB-MEA system (Multichannel Systems, DE). All the experiments were performed at 37°C in BPEL medium. MEA chambers were coated with fibronectin (40 μg/ml, Alfa Aesar, MA, USA) before seeding hiPSC-CMs. Acute dose–response curves were generated by adding aliquots at fixed 1:100 dilution every 10 min (Navarrete *et al*, 2013). Traces were analysed with a custom-made protocol to quantify both QT and RR intervals.

Chronic treatment with LUF7346 was evaluated using 24-well Multi-well MEA plates (Multichannel Systems, DE). Wells were coated with 75 μg/ml Matrigel before seeding hiPSC-CMs. The effects of chronic exposure to negative control (DMSO), positive control (5 μM E4031) and 3 μM LUF7346 were evaluated after 24-h and 48-h treatment. Measurements were performed at 37°C in BPEL medium. Traces were analysed with a custom-made protocol to quantify both QT and RR intervals.

### STV measurements

The variability of QT intervals was quantified, as previously described (Altomare *et al*, 2015; Pueyo *et al*, 2016), over 30 consecutive beats and expressed as STV, which indicates the average distance, perpendicular to the line of identity, in a Poincaré plot of $QT^n$ against $QT^{n+1}$. STV was calculated with the following equation:

$$STV = \sum_{n=2}^{30} \frac{|QT^n - QT^{n-1}|}{\sqrt{2} \times 29}$$

The effect of AST and AST+LUF7346 was expressed as delta percentage compared to the baseline value.

### Patch clamp

Electrical signals were recorded with an Axopatch 200B Amplifier (Molecular Devices, CA, USA) and digitised with a Digidata 1440A (Molecular Devices, CA, USA) connected to an x86 Windows PC running pClamp 10.4. All measurements were performed at 37°C. Data were analysed with ClampFit 10.4 (Molecular Devices, CA, USA), OriginPro 8.5 (OriginLab Corp, MA, USA) and Prism 7.0a (GraphPad Software, CA, USA) for Mac.

### Voltage clamp

Voltage-clamp studies were performed in the whole-cell configuration. Cells were superfused with a Tyrode's solution containing (mM): 154 NaCl, 5.4 KCl, 1.8 CaCl$_2$, 1 MgCl$_2$, 5 HEPES-NaOH, 5.5 D-Glucose; pH was adjusted to 7.35 with NaOH. L-type Ca$^{2+}$ current (I$_{CaL}$) and the slow delayed rectifier K$^+$ current (I$_{Ks}$) were blocked, respectively, by adding 5 μM nifedipine (Sigma-Aldrich, MO, USA) and 1 μM JNJ-303 (Tocris Biosciences, UK) to the extracellular solution.

Glass capillaries (1.5–3 MΩ) were filled with an intracellular solution containing (mM): 125 K-gluconate, 20 KCl, 10 NaCl, 10 HEPES, 5 K$_2$-ATP; pH was adjusted to 7.2 with KOH. Cell capacitance and series resistance were calculated in each cell and compensated from 65 to 80% to maintain the error on the superimposed voltage below 5 mV (average value: 3.05 ± 0.34 mV). hERG steady-state activation was evaluated with a standard two-pulse protocol (Rosati *et al*, 1998), with 4 s voltage steps from −60 mV to +30 mV, every 10 mV, followed by 16 s at −40 mV. Holding potential was set to −40 mV. Current/voltage (I/V) relationships were obtained by measuring the peak of the tail current at −40 mV and plotting it as a function of the conditioning voltage. The steady-state activation curve was generated by normalising tail currents at −40 mV to the maximal tail current (Liu & Trudeau, 2015). hERG steady-state inactivation was evaluated with a standard three-pulse protocol (Gianulis & Trudeau, 2011). From a holding potential of −80 mV, channels were fully activated and rapidly inactivated with a 500-ms step to +20 mV. The inactivation on previously activated channels was removed by 5-ms steps from −110 mV to +30 mV. Instantaneous currents were subsequently measured at +20 mV. To generate steady-state inactivation curves, the instantaneous current at +20 mV was normalised by the maximal instantaneous current. Both steady-state activation and inactivation curves were fitted with the following Boltzmann's function:

$$y = \frac{A_1 - A_2}{1 + e^{(x-x_0)/dx}} + A_2$$

hERG voltage-dependent deactivation was evaluated with a standard two-pulse protocol (Liu & Trudeau, 2015): I$_{Kr}$ was evoked by

voltage steps to +20 mV followed by voltage steps from −120 mV to 0 mV, every 20 mV. Tail currents were fitted with the following biexponential function:

$$y = y_0 + A_1 e^{(-x/\tau_1)} + A_2 e^{(-x/\tau_2)}$$

LUF7346 was tested at 3 μM for HEK293 hERG and 5 μM for hiPSC-CMs. E-4031 5 μM (Tocris Bioscience, UK) was used to ensure a complete blockade of hERG and allow the subtraction of background currents.

$I_{Ks}$ was measured in WT hiPSC-CMs with a Tyrode's solution containing: 5 μM E4031 (Tocris Biosciences, UK) to block $I_{Kr}$, 5 μM nifedipine (Sigma-Aldrich, USA) to block L-type $Ca^{2+}$ current and isolated with 1 μM JNJ-303 (Tocris Bioscience, UK). A 3-step voltage-clamp protocol was used for the recordings. From a holding potential of −40 mV, three depolarisation steps of 3,750 ms to −40 mV, 0 mV and +40 mV were applied, followed by 4,250 ms at −40 mV. The cycle length was 10 s.

$I_{CaL}$ was isolated in WT hiPSC-CMs with a modified Tyrode's solution containing (mM): 154 tetraethylammonium chloride, 5 $CaCl_2$, 1 $MgCl_2$, 5 HEPES, 5.5 D-glucose; pH was set to 7.4 with CsOH. 2 mM 4-aminopyridine (4-AP) and 0.01 mM tetrodotoxin (TTX) were added to block $K_v4.3$ and TTX-sensitive sodium channels (mainly $Na_v1.5$), respectively; pH was recalibrated to 7.4 with HCl after 4-AP addition. Pipette solution contained (mM): 115 CsCl, 20 tetraethylammonium chloride, 0.5 $MgCl_2$, 10 EGTA, 5 HEPES, 0.4 GTP-Tris salt, 5 ATP-Mg salt, 5 phosphocreatine-Mg salt; pH was set to 7.2 with CsOH. $I_{CaL}$ was evoked with a 4-step protocol as previously described (Rocchetti *et al*, 2014): 15 ms at holding potential (−80 mV), 1,000 ms of pre-step at −50 mV, 14 steps of 300 ms from −60 mV, every 10 mV and 65 ms at holding potential. Cycle length was 10 s. Sampling rate was 10 kHz, and traces were filtered with a low-pass Bessel filter at 2 kHz.

### Current clamp

Current-clamp experiments were performed in the perforated patch configuration. Cells were superfused with a Tyrode's solution containing (mM): 154 NaCl, 5.4 KCl, 1.8 $CaCl_2$, 1 $MgCl_2$, 5 HEPES-NaOH, 5.5 D-glucose; pH was adjusted to 7.35 with NaOH.

Glass capillaries (2–3.5 MΩ) were filled with an intracellular solution containing (mM): 125 K-gluconate, 20 KCl, 10 NaCl, 10 HEPES; pH was adjusted to 7.2 with KOH. Amphotericin B (Sigma-Aldrich, MO, USA) was dissolved in DMSO just before the experiments and added to the intracellular solution to reach a final concentration of 0.22 mM. LUF7346 was used at 1, 3, 5, 10 μM concentrations.

### Quantitative reverse-transcription PCR (RT–qPCR)

Total RNA was isolated using the NucleoSpinRNA kit (Macherey-nagel), following the manufacturer's instructions. Eight hundred nanograms of RNA was reverse-transcribed using the iScriptTM-cDNA Synthesis kit (Bio-Rad). Expression profiles of genes of interest were determined using the iQTM Universal SYBR Green Supermix (Bio-Rad). Gene expression was analysed using the $\Delta C_t$ method where the gene of interest was compared to the housekeeping gene *RPL37A*. Primers used were previously described: *TNNT2*,

### The paper explained

#### Problem
The long-QT syndrome (LQTS) is an arrhythmogenic disorder of the heart, with a prevalence of 1:2,000 newborns, which may lead to sudden cardiac death on the onset of the first arrhythmogenic event. In recent years, both diagnosis and (to a certain extent) therapies for LQTS have improved; however, there remains a need for new pharmacological approaches. In the process of drug development, predicting the toxicity and the efficacy of novel compounds represents a remarkable challenge, with a combination of *in vitro*, *ex vivo* and *in vivo* tests, with animal studies representing a substantial portion. Nevertheless, a consistent number of drugs are (i) rejected because of lack of proper *in vivo* effect or, worse, (ii) withdrawn from the market because they induce severe arrhythmias, especially in the presence of more susceptible genetic backgrounds. A better tool to improve the predictivity of these aspects is then required.

#### Results
As experimental model, we used cardiomyocytes derived from induced pluripotent stem cells (hiPSC-CMs) from patients with LQTS showing reduced cardiac K⁺ currents. We also tested their genetically engineered isogenic pairs and an unrelated healthy control line. We identified a small molecule, LUF7346 that successfully rescued all the genetic forms of LQTS tested. LUF7346 acted by modulating the activity of the cardiac hERG potassium channel. This small molecule also corrected the drug-induced form of LQTS in the presence of both wild-type or LQTS genetic background. Patient-specific, isogenic hiPSC-CMs proved to be a powerful tool for drug screening and safety pharmacology.

#### Impact
Implementation of isogenic, patient-specific hiPSC-CMs in drug screening and testing will likely increase the predictivity in the identification of not only drug efficacy, but also proarrhythmic effects, thus helping the drug development process. Furthermore, hERG allosteric modulation proved promising for the treatment of congenital and drug-induced LQTS *in vitro*.

*KCNH2-1a*, *KCNE2*, *KCNQ1*, *KCNJ12* and *CACNA1C* (Bellin *et al*, 2013); *RPL37A* and *KCNH2-1a/1b* (Zhang *et al*, 2014); *HCN4* (Moretti *et al*, 2010); *SCN5A* (Birket *et al*, 2015).

### Movies

Movies of beating monolayers were acquired with a Nikon DS-2MBW camera connected to a Nikon Eclipse Ti-S microscope, controlled by the Nikon NIS-Element BR software. Lens magnification was 10× with a PhL contrast filter.

### Statistics

Student's *t*-test, one-way or two-way ANOVA for paired or unpaired measurements were applied as appropriate to test for differences in means between groups/conditions. Kruskal–Wallis test and two-tailed Wilcoxon signed-rank test were used when the normality assumption did not hold. Post hoc comparison between individual means or medians was performed by Tukey's method, and *P*-values have been corrected for multiple testing using the Holm–Sidak or Dunn's method. Detailed statistics and exact *P*-values are indicated in each figure legend. Data are expressed and plotted as the mean ± SEM. Statistical significance was defined as $P < 0.05$

(n.s., not significant). Exact *P*-values for each comparison are expressed in data information within the respective figure legend. The sample size for each experiment is specified in the respective figure legend. Statistical analyses were performed with GraphPad Prism 7.0a for Mac and R 3.2.4 (The R Foundation for Statistical Computing).

**Expanded View** for this article is available online.

## Acknowledgements

We thank M. Rocchetti and M. Lecchi (both at the University of Milano-Bicocca, Italy) and E. Ficker (University of Cleveland, USA) for kindly providing HEK293 hERG cells used for electrophysiology and for the binding assays, respectively; J. Stępniewski (Jagiellonian University, Kraków) for help with LUF7346 chronic treatment experiments; S. Tsonaka (Leiden University Medical Center) for statistical assistance. This work was supported by the following grants: CVON (HUSTCARE): the Netherlands CardioVascular Research Initiative (the Dutch Heart Foundation, Dutch Federation of University Medical Centres, the Netherlands Organisation for Health Research and Development and the Royal Netherlands Academy of Sciences) (CLM and LS); The Netherlands Institute of Regenerative Medicine (NIRM) (CLM and MB); the European Research Council (ERCAdG 323182 STEMCARDIOVASC) (CLM and MB); the Chinese Scholarship Council (ZY).

## Author contributions

LS performed single cell and MEA electrophysiology measurements and analysis, as well as RT-qPCR analysis. ZY performed LUF compounds *in vitro* binding assays. DW-vO performed cell culture, MEA preparations and RT-qPCR. JPDvV performed LUF compounds synthesis. AM and K-LL provided LQT1$^{R190Q}$ and LQT2$^{N996I}$ hiPSCs. MB performed MEA measurements. LS, API, CLM and MB were involved in experimental design. LS, CLM and MB wrote the manuscript.

## Conflict of interest

C.L.M. is co-founder and advisor of Pluriomics.

## For more information

LQT1: http://www.omim.org/entry/192500
JLNS: http://www.omim.org/entry/220400
LQT2: http://www.omim.org/entry/613688

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
