## [Review Process File · EMBO Molecular Medicine]

A new hERG allosteric modulator rescues genetic and drug-induced Long-QT Syndrome phenotypes in cardiomyocytes from isogenic pairs of patient induced pluripotent stem cells

Luca Sala, Zhiyi Yu, Dorien Ward-van Oostwaard, Jacobus P.D. van Veldhoven, Alessandra Moretti, Karl-Ludwig Laugwitz, Christine L. Mummery, Adriaan P. IJzerman, Milena Bellin

Corresponding author: Milena Bellin, Leiden University Medical Center

Review timeline:

Submission date:	29 January 2016
Editorial Decision:	26 February 2016
Revision date:	26 May 2016
Editorial Decision:	13 June 2016
Revision date:	24 June 2016
Accepted:	29 June 2016

Transaction Report:

Editor: Céline Carret

1st Editorial Decision

26 February 2016

Thank you for the submission of your manuscript to EMBO Molecular Medicine. We have now heard back from the three referees whom we asked to evaluate your manuscript. Although the referees find the study to be of potential interest, they also raise a number of concerns that must be addressed in the next final version of your article.

You will see from the reports below that there are a few issues that would need to be satisfactorily addressed for the paper to be further evaluated in EMBO Mol Med, especially as referees 2 and 3 are concerned about novelty, the putative translational implications must be strengthened as suggested by both referees 1 and 3.

Given these evaluations, I would like to give you the opportunity to revise your manuscript, with the understanding that the referees' concerns must be fully addressed and that acceptance of the manuscript would entail a second round of review. Please note that it is EMBO Molecular Medicine policy to allow only a single round of revision and that, as acceptance or rejection of the manuscript will depend on another round of review, your responses should be as complete as possible.

Please read below for important editorial formatting.

I look forward to receiving your revised manuscript.

***** Reviewer's comments *****

Referee #1 (Remarks):

Sala et al. use three isogenic pairs of iPSC-CMs to study the effects of a novel allosteric activator of the hERG channel. Building on their prior work to identify novel hERG allosteric modulators (Yu et al. 2015), the authors test the most promising lead compound, LUF7346, on three patient-derived iPSC-CM lines and a genome-edited matching pair line, as well as an unrelated wild-type control line. The major conclusions reached by the authors are (1) LUF7346 activates hERG in iPSC-CMs and corrects QT duration in genetic and acquired forms of long QT syndrome (LQTS), and (2) isogenic sets of patient-derived iPSC-CMs are important for drug discovery in disease and safety pharmacology. The efficacy data for LUF7346 in normalizing repolarization is solid, but the study lacks evaluation of off-target effects on non-hERG currents in iPSC-CMs. While the use of isogenic lines makes intuitive sense, the data actually do not illustrate the importance of the using isogenic lines. To do so, the authors should show how their conclusions would be altered in the absence of isogenic lines.

Major points:

1. The authors should provide quality control data to show that the iPSC lines are equivalent and differentiate into cardiomyocytes with equivalent efficiency and expression of key relevant cardiac genes/channels/currents. At the minimum, gene expression of the components of IKr should be shown.
2. The spontaneous beat rate of the cells is low (~30 bpm) and variable between lines. It appears that pacing the isolated cells or the sheets of cells on MEAs was problematic. If it is not possible to pace the cells to make their rates the same, then efforts should be made to correct the QT interval to the beat rate, by Bazett's formula.
3. Can the authors provide data that support the specificity of LUF7346, e.g. lack of effect on INa, IK1, or IKs.
4. The authors point to their work validating and highlighting the importance of using isogenic lines. However, the data do not actually explicitly show the advantage of isogenic lines as they do not show that any of the key findings are contingent upon using isogenic lines. In some passages they allude to differences that they observe and suggest that these point to the need for isogenic lines, but these are not convincing. For instance, the wild-type iPSC-CMs have shorter QT duration than any of the corrected patient-derived lines is mentioned, but how does this test the hypothesis that it is important to use isogenic lines? The authors also refer to Fig. EV4 as supporting this point but it compares a mutant compared to isogenic control genotype and so is not pertinent for the point being discussed.
5. To follow up point #4, what is the variation in QT duration between different clones with identical genotype? The authors needs to show that the variance between clones within genotype is less than the difference between between clones between genotypes (e.g. wild-type compared to LQT2-corrected for several different clones).

Minor points:

1. The QT of R190Q is said to be significantly longer than any of the other lines including R594Q/R594Q (JLNS). This is rather surprising. Is this difference also seen on the patient's EKGs? How do the authors explain this observation?

2. Many cases of acquired LQTS are due to drug interaction with channel polymorphisms. The authors do not investigate this more common type of acquired LQTS and might mention this in the discussion.
3. The presentation of the data in Figures 3A-B and 4A-B with multiple overlaid traces and error bars could be improved.
4. The recordings in figure 2C appear to be noisy and inappropriately filtered. There are no activation or deactivation curves measured for the individual lines.
5. In Fig. 5, EADs and DADs should be quantified and summarized across many cells and biological replicates. Current clamp recordings could be done as another measure of the arrhythmogenic potential of the mutations and the ability of LUF7346 to reverse that phenotype.
6. Specific statistical analysis used for each figure should be mentioned if different from a standard student's t-test.
7. Discussion is rather lengthy.

Referee #2 (Comments on Novelty/Model System):

The technical quality, adequacy of systems and impact are all high; however, many of the systems and tested treatment, methodology are based upon previously published data, reducing the novelty. Yet we still believe that this manuscript is worthy of publication.

Referee #2 (Remarks):

In the manuscript by Sala et al., allosteric modulation of hERG using LUF7346 demonstrates a treatment effect in congenital, acquired and the combination of the two in a long QT syndrome phenotype modeled hiPSC-CMs. They also demonstrate that isogenic pairs of hiPSC-CMs are a valuable platform for drug screening and pharmacologic safety in the complexity of genetic background. The manuscript is written well, the science is sound, and the results and limitations are concise. We suggest some minor revisions:

Minor Revisions:

1. Please correct QT intervals with the relatively long RR and compare QTc in fig 2B. What are the normal range of QT and QTc in these iPSC-CMs? Can $QT < 0.4$ sec and $QTc < 0.44$ sec be applied?
2. Please consider modifying the color in Fig. 3A (right) and Fig. 4A (right) for better visualization.
3. The discussion section seems somewhat lengthy with partial repetition of methods and should be shortened accordingly.
4. We disagree with the statement "The main risk of treating LQTS with hERG activators is excessive shortening of the QT interval, which may result in arrhythmic events" (line 341) as short QT syndrome rarely cause arrhythmic events, and the prolonged RR interval or cessation of spontaneous beating (line 179) should be more catastrophic. Please revise the statement.

Referee #3 (Comments on Novelty/Model System):

Authors describe effects of an new hERG channel activator on hiPSC-CM. Such compounds are not completely new and it remains unclear what could be the advantage of that new compound. Authors ignore work with comparable compound done in sophisticated transgenic rabbits. Therefore it is unclear why it is so important to use hiPSC-CM. Authors interpret different effect size on APD in

different cell lines. However, a meaningful interpretation needs data about compound effects on hERG in different cell lines.

Referee #3 (Remarks):

Authors address the ability to rescue genetic and drug-induced Long-QT syndrome in humans by pharmacological modulation of hERG gating. They do so by measuring hERG currents and action potentials in isolated hPSC-CM. In addition, field potentials were measured in cultured cell using a multielectrode array. For the experiments patient specific cell lines (for LQT1 and LQT2) including respective isogenic control were employed. In addition, data are given for an independent control line. The idea on pharmacological activation focus interests for more than years. Authors are excellent experts in the field of hPSC-CM electrophysiology. Recently some of the authors reported reversal of QT-prolongation by another hERG channel modulator (ML-T531) in a different model of mutated KCNQ1 channels (Zhang H et al. *Proc Natl Acad Sci U S A*. 2012 Jul 17;109(29):11866-71). Therefore, present work is a somewhat a logical extension of previous work. In the present work, authors used a new hERG modulator, however new insights are limited because of methodological shortcomings.

Major points

1. New and allosteric modulator

The field of hERG modulators (most of them used as activators) is rapidly growing and there is clearly a need for classification. Most of them are classified as allosteric compounds, since they bind far from the binding site of classic hERG blockers. Another classifications (so called typ 1 vs. typ 2 activators, depending on the mechanism of current activation) may more helpful to predict functional consequences of hERG modulation (for review see: Perry M, Sanguinetti M, Mitcheson J. Revealing the structural basis of action of hERG potassium channel activators and blockers. *J Physiol*. 2010 Sep 1;588 (Pt 17):3157-67. doi: 10.1113). Authors use LUF7346 as a new and allosteric compound, but do not compare LUF7346 effects to effects evoked by older/non-allosteric modulators. This fact represents a major limitation with respect to interpretations and conclusions. Therefore, it is hard to estimate, what impact the results presented in the manuscript, will have for further drug development.

2. Reversal of QT-prolongation vs. antiarrhythmic activity

One of the oldest hERG activators was NS1643 reported 10 years ago (Hansen RS et al. *Mol Pharmacol*. 2006 Jan;69(1):266-77). Discovery of NS 1643 evoked an enormous interest and consequently that compound was employed in sophisticated transgenic rabbit models of long QT. As described for LUF7346 in the present manuscript the old compound NS1643 could revert APD-prolongation and was successful in rescuing drug-induced long-QT syndrom (Diness TG et al. *Cardiovasc Res*. 2008 Jul 1;79 (1):61-9.), but increased arrhythmias in a Langendorff-model of genetic long QT (Bentzen BH et al. *J Cardiovasc Pharmacol*. 2011 Feb;57 (2):223-30). Collectively, this results clearly illustrates that reversal of APD and or QT-prolongation alone cannot be as a surrogate parameter to predict effective treatment of long-QT.

Minor points

1. QT interval under basal conditions

It is one of the strongest points of the present manuscript to show QT intervals of different cell lines including their respective controls under basal conditions. However at least from my point of view interpretation is rather tricky. In order to detect small but potentially harmful effects on QT in humans different rate algorithms were developed in order to adapt QT to actual heart rate. I would not expect that such formulations used in safety pharmacology could be simply transferred to hPSC-CM. However, I would expect longer QT at slower rate. LQT1 cells beats almost two times slower, but QT is only slightly increased? Furthermore, why is QT not different in LQT2N961 vs. LQT2corr?

2. Findings with astemizol and personalized medicine

Authors argue the proposed measurements in hPSC-CM could be useful as a tool to assess risk for QT-prolongation in the context of personalized medicine. I strictly disagree. From the today perspective astemizol is just an ordinary hERG blocker (Zhou Z et al. *J Cardiovasc Electrophysiol*. 1999 Jun;10(6):836-43). The disastrous incidence of torsade de points arrhythmias and sudden death leading to withdrawn from the market was rather the result of (at that time unknown) pharmacokinetic interaction on the level of cytochromes than peculiar pharmacodynamics that could be elucidate only by employing hPSC-CM.

3. Introduction: Trigger (p.3, l. 52)

Not only the acquired form of LQT is triggered by the factors listed? The congenital too?

4. Results: Same heterologous cells? (p. 5, l. 107)

Please clarify

5. Results: fast and slow component (p. 6, l. 126)

What is slow and fast component? Fast activation? Fast decay? Please be precise.

6. Interpretation of the amount of APD/QT-shortening by LUF7346 (p.15, l. 327)

Authors interpret different effect size in different cell lines. However a meaningful interpretation needs data about compound effects on hERG in different cell lines.

7. Comparison of LUF7346 effects on APD to effect of other hERG activators (p. 13, l. 287)

Authors compare LUF7346 effects on APD to effect of other hERG activators from the literature. This seems not justified, differences could be related to different systems.

8. Discussion in general

The discussion would profit from a clear structure.

9. Methods: voltage clamp compensation (p. 20, l. 459)

I am wondering that authors have used capacitance and series resistance compensation only when necessary. Please clarify

10. Proper management ("The paper explained") (p. 23, l. 525)

Authors report prevalence data of LQTS that can lead to sudden cardiac death if it is "not properly managed". This sentence is rather confusing. Do the authors already know what proper management could be?

11. Results: Table III (p. 37, l. 894)

In HEK293 cells expressing hERG authors report results of biexponential fits to decaying current traces. Obviously deactivation was slowed by LUF7346 and tracings could be fitted by one time constant only. However, where is the single tau for respective monoexponential fit at voltages >-80mV?

1st Revision

26 May 2016

Referee #1 (Remarks):

Sala et al. use three isogenic pairs of iPSC-CMs to study the effects of a novel allosteric activator of the hERG channel. Building on their prior work to identify novel hERG allosteric modulators (Yu et al. 2015), the authors test the most promising lead compound, LUF7346, on three patient-derived iPSC-CM lines and a genome-edited matching pair line, as well as an unrelated wild-type control line. The major conclusions reached by the authors are (1) LUF7346 activates hERG in iPSC-CMs and corrects QT duration in genetic and acquired forms of long QT syndrome (LQTS), and (2) isogenic sets of patient-derived iPSC-CMs are important for drug discovery in disease and safety pharmacology. The efficacy data for LUF7346 in normalizing repolarization is solid, but the study lacks evaluation of off-target effects on non-hERG currents in iPSC-CMs. While the use of isogenic lines makes intuitive sense, the data actually do not illustrate the importance of the using isogenic lines. To do so, the authors should show how their conclusions would be altered in the absence of isogenic lines.

Major points:

1. The authors should provide quality control data to show that the iPSC lines are equivalent and differentiate into cardiomyocytes with equivalent efficiency and expression of key relevant cardiac genes/channels/currents. At the minimum, gene expression of the components of IKr should be shown.

The point raised by the reviewer is extremely important. As reported in the literature, cardiac and non-cardiac differentiations efficiencies do vary a lot not only among distinct differentiation methods but also among independent (embryonic and induced) pluripotent stem cell lines (Osafune et al., Nature Biotech 2008; Denning et al., *Biochim Biophys Acta*, 2015). In particular, the variation among cardiac

differentiation methods is quite wide, with a range of 7% to >98% of cTnT⁺ cells (Talkhabi et al., *Life Sciences* 2016). However, although cardiomyocyte differentiation techniques were initially inefficient and not readily transferable across cell lines, there are now a number of more robust protocols available (reviewed in Mummery et al., *Circ Res*, 2012; BurrIDGE et al., *PLOS One*, 2011; Van den Berg et al., *Methods Mol Biol*, 2016). Here, we employed an identical monolayer differentiation protocol (Dambrot et al., *Exp Cell Res*, 2014) across all cell lines analysed, which resulted in contracting areas by day 8-10 from initiation of the differentiation in all cases. As an example, we have now provided representative movies of the beating monolayer at day 14 of differentiation (Source Data Movie 1-9).

Of note, although there is some variation in the number of generated cardiomyocytes among different experiment or different lines, the quality of the resulting cardiomyocytes was unchanged. It is important to note that this is a more general concept: although the efficiency of differentiation protocols has undergone a multifold increase over recent years as a result of culture condition optimization, this has not been paralleled by improvements in the electrophysiological properties of hPSC-CMs (upstroke velocity, resting membrane potential, ion channel expression remain low in comparison to adult CMs). This suggests that optimization has impacted quantitative rather than qualitative aspects of differentiation.

Importantly, the lines used here have been already described in original papers modelling LQTS and JLNS, where a detailed analysis of gene expression of the major ion channels shaping the AP were reported (Moretti et al. *New Engl J Med* 2010, Bellin et al. *EMBOJ* 2013, Zhang et al. *PNAS* 2014, Chen et al., *Eur H J in press*, doi:ehw189). To further show gene expression of I_{Kr} components under these same culture and differentiation conditions, we have now provided qPCR data for all the lines (Fig EV1B).

2. The spontaneous beat rate of the cells is low (~30 bpm) and variable between lines. It appears that pacing the isolated cells or the sheets of cells on MEAs was problematic. If it is not possible to pace the cells to make their rates the same, then efforts should be made to correct the QT interval to the beat rate, by Bazett's formula.

All the action potentials measured by patch clamp in isolated cells were recorded under stimulation at 1 Hz in all the tested lines. We apologize for not having clearly indicated this. We have now indicated the stimulation frequency in the legends of Figure 3, 4, and 5.

With regards to the MEA recordings, the referee is right in observing that we did not pace the clusters of cells. Just like the vast majority of published data using MEAs, spontaneous activity was recorded and analysed. Indeed, pacing of beating clusters or monolayers at MEA is technically difficult with MultiChannel System hardware for two main reasons: 1) the stimulus required by the clusters in order to be paced is far beyond both the amplifier capabilities (up to 5V, MultiChannel System STG-1001) and the durability of TiO electrodes, which were initially built for neuronal networks; 2) the stimulation applied to an electrode completely masks its signal, also affecting the neighbouring electrodes by the generation of large capacitive transients; these are always larger and longer than the Field Potential generated by the cluster. To the best of our knowledge, only one group managed to stimulate hiPSC-CMs clusters on MEA through a sophisticated, custom-made system (Kaneko et al., *Jap J Appl Phys*, 2012).

Since the QT interval is dependent on the heart rate, Bazett's and Fridericia's formulae have been clinically implemented to make QT values independent of RR values, thus allowing comparisons between QT at different beating frequencies as if they were recorded at 60 beats per minute; the positive relationship between RR- and QT intervals can be fitted with regression models. The steeper the relationship, the more changes in the RR affect QT values. QT correction methods aim at minimising the angular coefficient of this fitting.

When we fitted our raw data with a major-axis regression analysis, we obtained angular coefficients not different from 0, suggesting no QT-RR dependency and that the data should not be corrected (Figure EV2 – see below – and Table EV2). Since both Bazett's and Fridericia's corrections were implemented for RR intervals in the clinical range, these may then introduce artefacts for very high or very low frequencies.

Therefore, we believe that when we apply either Bazett's or Fridericia's formulae, we introduce an over-correction that would be misleading. For this reason we initially concluded that it was more informative to show the non-corrected FPD for each line and their corresponding RR interval. However, since this referee and the others highlighted this point, we have now added a bar graph representing also QTcB in

Figure 2B.

The specific effect that we see when we apply the major-axis regression model might be a characteristic typical of hiPSC-CMs and may require further experiments/discussion with data from a consistent number of cell lines. We thank the reviewer for raising this point.

[The unpublished data provided for the referees were removed]

3. Can the authors provide data that support the specificity of LUF7346, e.g. lack of effect on I_{Na} , I_{K1} , or I_{Ks} .

We thank the reviewer for raising this point. The effect of LUF7346 was indeed initially assessed on I_{Kr} and on the overall shape of the AP. By analysing the parameters of the AP in patched cells before and after addition of LUF7346, we observed no significant changes in the upstroke velocity, suggesting that most likely I_{Na} was not influenced by LUF7346, nor in the diastolic potentials, indicating that I_{K1} was also unaffected by the compound. The slight hyperpolarisation brought by the consistent increase in I_{Kr} also in diastole, and visible in qualitative AP Clamp experiments in Fig EV3D, is likely responsible for the trends in figures 3D, 4D and EV3C. However, to test whether LUF7346 has an effect on I_{Ks} or I_{CaL} , we measured these currents in WT hiPSC-CMs and confirmed that LUF7346 had no effect on these two currents (Fig EV3). These data contributed to further validation of the specificity of LUF7346.

4. The authors point to their work validating and highlighting the importance of using isogenic lines. However, the data do not actually explicitly show the advantage of isogenic lines as they do not show that any of the key findings are contingent upon using isogenic lines. In some passages they allude to differences that they observe and suggest that these point to the need for isogenic lines, but these are not convincing. For instance, the wild-type iPSC-CMs have shorter QT duration than any of the corrected patient-derived lines is mentioned, but how does this test the hypothesis that it is important to use isogenic lines? The authors also refer to Fig. EV4 as supporting this point but it compares a mutant compared to isogenic control genotype and so is not pertinent for the point being discussed.

We thank the referee for highlighting the point that we did not explicitly demonstrate the importance of using isogenic lines in the drug testing and as controls.

The main points supporting this concept can be summarised as follows:

1) The four wild-type lines used here show very different basal QT (or QTc) intervals (Fig EV1B). When screening the literature for identifying a “normal” QT-interval reference value for wild-type hPSC-CMs measured at $\sim 37^{\circ}\text{C}$, even bigger differences exist, with APD₉₀ varying from 122 ms to 645 ms (see Figure Referees 1, below).

[The unpublished data provided for the referees were removed]

2) The APD of some wild-type hPSC-CMs is longer than some LQTS-CMs. A wild type longer than an unrelated LQTS is an obvious problem for disease modelling, since it may mask the pathogenic phenotype. Our data clearly confirm this. For example, by randomly choosing LQT1^{corr} as control for JLNS^{R594Q} we would conclude that the homozygous *KCNQ1*^{R594Q} mutation is not pathogenic because it does not show any APD prolongation and there would be no need to search for treatments able to shorten the QT-interval as done here in this work.

3) Finally, in order to identify the drug concentration range that is necessary and sufficient to rescue the pathogenic phenotype, again a genetically matched control is essential. Indeed, the choice of a random control could lead to under/over estimation of the required dosage for rescuing the phenotype. For example, when analysing both LQT2^{N996I} and LQT1^{R190Q} by patch clamp (Fig 3C) we saw that a

concentration between 1 μM and 3 μM of LUF7346 was able to restore the control APD compared to their isogenic controls. However, by swapping the comparison, as could happen in the case of randomised controls (LQT1^{R190Q} vs LQT2^{corr} and LQT2^{N996I} vs LQT2^{corr}) a higher drug concentration would be needed in the first comparison, while even not necessary in the second comparison. These conclusions are relevant especially in the effort to implement patient-specific hiPSC-CMs in the Precision Medicine Initiative.

We have now clarified this concept by including this point in the discussion (page 14, lines 306-309) and by showing a direct comparison with additional patch clamp data on the LQT1^{corr/R190Q} isogenic pair at two different drug concentrations.

5. To follow up point #4, what is the variation in QT duration between different clones with identical genotype? The authors needs to show that the variance between clones within genotype is less than the difference between clones between genotypes (e.g. wild-type compared to LQT2-corrected for several different clones).

Whilst we appreciate the point, the analysis of the variation of QT and AP duration between different hiPSC clones is beyond the scope of this work. Furthermore, the genetic modification of the hiPSCs lines used here was achieved by using a classical gene targeting approach based on homologous recombination with the use of replacement vectors containing very long homology arms, for which targeting efficiency is notoriously low (Nieminen et al. *Exp Cell Res*, 2010). Consequently, usually one single targeted clone is selected and fully characterised. However, we appreciate the point of the referee and here below we report what has been shown in terms of APD₉₀ variation for three independent clones with R190Q *KCNQ1* genotype from two individuals (PII-2 and PIII-2) and three independent clones with wild-type genotype (C-1 and C-2) (from Supplementary Figure 7 in Moretti et al. *New Engl J Med*, 2010). This graph indicates that the variation between clones with the same genotype is indeed less than between clones with different genotypes.

[The unpublished data provided for the referees were removed]

Minor points:

1. The QT of R190Q is said to be significantly longer than any of the other lines including R594Q/R594Q (JLNS). This is rather surprising. Is this difference also seen on the patient's EKGs? How do the authors explain this observation?

We agree with the referee that this was a quite surprising result. The LQT1^{R190Q} was derived from a male LQT1 patient with QTc of 462 ms (Moretti et al. *New Engl J Med*, 2010). The JLNS^{R594Q}, instead, does not have a corresponding patient with identical genotype, since it was generated by gene targeting in the LQT1^{R594Q} line. The latter was derived from a female LQT1 patient with a QTc of 506 ms (Zhang et al, *PNAS*, 2014). Finally, the LQT2^{N996I} line was derived from a female patient with a QTc of 617 ms, Bellin et al., *EMBOJ*, 2013). From these data (and from a thorough analysis of the literature, see Figure Referees 3, below) we conclude that there is likely no unequivocal relationship between the APD/FPD measured *in vitro* in hiPSC-CMs and the ECG in the corresponding patients.

[The unpublished data provided for the referees were removed]

Importantly, the LQTS is an autonomous disease of the cardiomyocytes but we should keep in mind that isolated cells *in vitro* may not fully recapitulate the higher complexity of a whole 3D organ, including sympathetic activity, the effects of autocrine/paracrine molecules and the hormonal landscape of one individual. Overall, we believe that these observations emphasize even more the importance of using isogenic hiPSC-CMs.

To the best of our knowledge, no direct correlation has been made so far between clinical QTc data and

in vitro hiPSC-CMs APD/FPD, but we do agree that once reached for a consistent number of lines, it would be worth analysing the variables to possibly extrapolate the sources of these variations.

We thank the referee for raising this point that warrants further follow up.

2. Many cases of acquired LQTS are due to drug interaction with channel polymorphisms. The authors do not investigate this more common type of acquired LQTS and might mention this in the discussion.

We thank the referee for this comment. We agree that this is an important point and we have mentioned it in the Introduction (page 3, line 57-60).

3. The presentation of the data in Figures 3A-B and 4A-B with multiple overlaid traces and error bars could be improved.

We agree with the referee that showing data for many lines in one unique graph requires a clear graphical representation. Since we believe that showing one graph is more informative than splitting the chart in several sub-groups, we provide an alternative visualisation of the results for Figures 3A-B and 4A-B. Panels A have been simplified with the presence of vertical lines to help the reader in evaluating the T peak; panels B are represented with a heatmap, in which the colour code defines the magnitude of the effect. We leave the editor the decision on which version of the figure best represent the data. An option may be to include the heatmap in the main text and the original representation in the supplementary data.

4. The recordings in figure 2C appear to be noisy and inappropriately filtered. There are no activation or deactivation curves measured for the individual lines.

We agree with the referee and we have now improved the quality of the current traces shown in Figure 2C by choosing a lower weight of the line. The filter applied on the Multiclamp 200B amplifier was a Lowpass Bessel Filter at 2 kHz, which is considered appropriate for I_{K_r} measurements.

The point of Figure 2C-E was to evaluate the effect of LUF7346 on wild-type hERG channels, in the context of a more physiologically relevant human cell system than overexpression models. All the hiPSC lines have only wild-type hERG channel expressed on the membrane (the N996I-KCNH2 mutation displayed a trafficking defect, Bellin et al. *EMBOJ*, 2013), therefore we chose one representative line in which to measure activation and deactivation. Measuring the compound effect in all these lines would not provide more information.

5. In Fig. 5, EADs and DADs should be quantified and summarized across many cells and biological replicates. Current clamp recordings could be done as another measure of the arrhythmogenic potential of the mutations and the ability of LUF7346 to reverse that phenotype.

We agree with the referee that the arrhythmic effect should be quantified. In this study DADs were never detected, while EADs were detected only in the JLNS^{R594Q}-CMs in the presence of AST during current clamp recordings. Quantification of EADs is described in the text of the manuscript on page 11 that now reads: “AST also induced arrhythmic events in JLNS^{R594Q}-CMs (in 20.6% of the cells)”

By following the referee’s comment, we have now conducted a deeper analysis of the arrhythmogenic risk by examining the short-term variability of the repolarisation phase (STV) of FP at the MEA (Fig 5B-C). This analysis has never been previously applied to hiPSC-CMs, and was revealed as being fairly predictive of the arrhythmogenic risk in these isogenic pairs.

6. Specific statistical analysis used for each figure should be mentioned if different from a standard

student's t-test.

We agree with the referee and we have now specified in the legend of each figure which test was applied, with exact p-values for all the comparisons that reached statistical significance.

7. Discussion is rather lengthy.

We agree with the referee. We have shortened and reorganised the Discussion, which is now more focussed.

Referee #2 (Comments on Novelty/Model System):

The technical quality, adequacy of systems and impact are all high; however, many of the systems and tested treatment, methodology are based upon previously published data, reducing the novelty. Yet we still believe that this manuscript is worthy of publication.

Referee #2 (Remarks):

In the manuscript by Sala et al., allosteric modulation of hERG using LUF7346 demonstrates a treatment effect in congenital, acquired and the combination of the two in a long QT syndrome phenotype modeled hiPSC -CMs. They also demonstrate that isogenic pairs of hiPSC-CMs are a valuable platform for drug screening and pharmacologic safety in the complexity of genetic background. The manuscript is written well, the science is sound, and the results and limitations are concise. We suggest some minor revisions:

We thank the reviewer for appreciating the manuscript and for the comments.

Minor Revisions:**1. Please correct QT intervals with the relatively long RR and compare QTc in fig 2B.**

What are the normal range of QT and QTc in these iPSC-CMs? Can QT < 0.4 sec and QTc < 0.44 sec be applied?

As requested by this and the other referees (see answers to Ref #1 and #3), we have now corrected the QT interval using Bazett's formula and we show these results in Figure 2B (bottom). However, we believe that in the specific case of the measurements presented in this work, it is not necessary to apply a QT correction (when plotting QT vs RR, the linear coefficient is not significantly different from zero, Fig. EV2); rather, both Bazett's and Fredericia's formulae introduce an overcompensation (linear coefficient significantly different from zero, Fig. EV2). In the view of our results, we feel we should make a cautionary note in the application of rate-correction to *in vitro* data (spontaneous action and field potentials): Bazett's formula was developed for rates within the clinical range, therefore when the beating rate is e.g. <0.5 Hz (< 30 b/min) Bazett's correction may be unreliable. We have added this comment in the discussion, page 13, line 294.

Both the QT and QTc ranges of wild-type hiPSCs lines were quite wide, ranging from ~150 ms (QT of WT hiPSC-CMs) to ~400 ms (QT of LQT1^{cor} hiPSC-CMs); similarly, the range of the lines carrying mutations in either *KCNQ1* or *KCNH2* was wide, ranging from ~200 ms (LQT1^{R594Q}) to ~600 ms

(LQT1R^{190Q}). Therefore, we believe that QT <0.4 s and QTc <0.44 s cannot be applied to hiPSC-CMs. As confirmation, a thorough analysis of all the published data so far clearly demonstrate an absence of direct correlation between QT interval measured in patients' ECG (i.e. in whole heart) and AP or FP measured *in vitro* in hiPSC-CMs (see Figure Referees 3 above – answer to ref#1, point 4). Speculative explanations of these differences include lack of a 3D system *in vitro*, compared to whole heart, immaturity of hiPSC-CMs compared to adult CMs, but also differences between the physiological environment of a whole heart (where other cell types are present and hormones are present in the circulation) that are currently not recapitulated in *in vitro* cultures. Further research is warranted to interpret these differences.

2. Please consider modifying the color in Fig. 3A (right) and Fig. 4A (right) for better visualisation.

As suggested by this and other referees, we have represented these results as a heatmap, for better visualisation. We leave it to the editor to decide which version of the figure best represent the data.

3. The discussion section seems somewhat lengthy with partial repetition of methods and should be shortened accordingly.

We agree with the referee. We have shortened and reorganised the Discussion, which is now more focussed.

4. We disagree with the statement "The main risk of treating LQTS with hERG activators is excessive shortening of the QT interval, which may result in arrhythmic events" (line 341) as short QT syndrome rarely cause arrhythmic events, and the prolonged RR interval or cessation of spontaneous beating (line 179) should be more catastrophic. Please revise the statement.

We thank the referee for this insightful comment. We have now modified this statement in the discussion (page 16, lines 373-375) which now reads: "[...] prolonged RR interval and cessation of spontaneous beating as detected in our assays might result in more severe *in vivo* effects, thus forming an obstacle clinical translation."

Referee #3 (Comments on Novelty/Model System):

Authors describe effects of an new hERG channel activator on hiPSC-CM. Such compounds are not completely new and it remains unclear what could be the advantage of that new compound. Authors ignore work with comparable compound done in sophisticated transgenic rabbits. Therefore it is unclear why it is so important to use hiPSC-CM. Authors interpret different effect size on APD in different cell lines. However, a meaningful interpretation needs data about compound effects on hERG in different cell lines.

We thank the referee for the constructive criticism. Although LUF7346 is a novel compound, we are aware that this class of hERG channel activators is not new and it has been tested for the correction of LQTS in independent studies in heterologous systems, hiPSC-CMs, and sophisticated transgenic rabbit models (to which we refer in the Introduction, lines 65-67). The advantage of the new LUF7346 molecule resides in 1) its proven allosteric modulation mechanism of action and 2) its potency, which is higher than for previously reported hERG activators (as we discuss in lines 359-362). In the revised paper we have also directly compared LUF7346 with known hERG modulators (Fig EV4). The importance of demonstrating that hiPSC-CMs are a useful and reliable tool to test these compounds is

beneficial mainly for three reasons: 1) these cells represent an infinite source of cardiomyocytes (therefore suitable to the implementation in the drug screening and drug safety processes) 2) they represent a physiologically relevant human system, expressing the main ion channels involved in the AP generation; and 3) they can contribute to the reduction (but not replacement) of animal use in the research. In conclusion, we do not claim that hiPSC-CMs can replace existing models, but we do demonstrate their value in early drug discovery and safety pharmacology, with the possibility of testing drugs on patient-derived samples, with isogenically-matched controls at early phases of the preclinical research.

Finally, we respectfully disagree with the referee that meaningful interpretation needs data on compound effect on hERG in different cell lines. The mechanism of action of the small molecule LUF7346 has been thoroughly characterised in wild-type hERG channels (radioligand binding assays in HEK hERG cells, I_{Kr} measurements in HEK hERG, and I_{Kr} measurements in wild-type hiPSC-CMs; see also Yu et al., *Eur J Med Chem*, 2015 for a more detailed characterisation of LUF7346 allosteric modulation), as was done for all previously described activators, because this is the channel that is expressed in the general population and for which it is important that the molecule is active. Then, testing on different LQTS genetic backgrounds is meaningful with regards to the impact on the APD/FPD and arrhythmic phenotypes.

Referee #3 (Remarks):

Authors address the ability to rescue genetic and drug-induced Long-QT syndrome in humans by pharmacological modulation of hERG gating. They do so by measuring hERG currents and action potentials in isolated hiPSC-CM. In addition, field potentials were measured in cultured cell using a multielectrode array. For the experiments patient specific cell lines (for LQT1 and LQT2) including respective isogenic control were employed. In addition, data are given for an independent control line. The idea on pharmacological activation focus interests for more than years. Authors are excellent experts in the field of hiPSC-CM electrophysiology. Recently some of the authors reported reversal of QT-prolongation by another hERG channel modulator (ML-T531) in a different model of mutated KCNQ1 channels (Zhang H et al. Proc Natl Acad Sci U S A. 2012 Jul 17;109(29):11866-71). Therefore, present work is a somewhat a logical extension of previous work. In the present work, authors used a new hERG modulator, however new insights are limited because of methodological shortcomings.

Major points

1. New and allosteric modulator

The field of hERG modulators (most of them used as activators) is rapidly growing and there is clearly a need for classification. Most of them are classified as allosteric compounds, since they bind far from the binding site of classic hERG blockers. Another classifications (so called typ 1 vs. typ 2 activators, depending on the mechanism of current activation) may more helpful to predict functional consequences of hERG modulation (for review see: Perry M, Sanguinetti M, Mitcheson J. Revealing the structural basis of action of hERG potassium channel activators and blockers. J Physiol. 2010 Sep 1;588 (Pt 17):3157-67. doi: 10.1113). Authors use LUF7346 as a new and allosteric compound, but do not compare LUF7346 effects to effects evoked by older/non-allosteric modulators. This fact represents a major limitation with respect to interpretations and conclusions. Therefore, it is hard to estimate, what impact the results presented in the manuscript, will have for further drug development.

It is correct that most of our negative allosteric modulators might be termed as hERG activators as well. However, we found several compounds (7f2 and 7f4) with a similar chemical scaffold to LUF7346 that decreased the dissociation of [³H]dofetilide, indicating that they might act as positive allosteric modulators in our previous publication (Yu Z, *Eur. J Med Chem*, 2015). Accordingly, although the concept of activators is helpful to predict and describe functional consequences of hERG modulation, it excludes the possibility of positive allosteric modulators or certain allosteric modulators without functional impacts (so-called silent allosteric modulators or neutral allosteric ligands). Therefore, the terminology of allosteric modulators is more comprehensive and appropriate compared to that of activators, and this new concept in the research field of hERG channels is strongly suggested to be more

widely used in follow-up studies. Much of the above is reflected in a recent authoritative review in *Pharmacological Reviews* in which preferred terminology is proposed in Table 1 (Christopoulos et al., *Pharmacol Rev*, 2014). Nevertheless, for clarity we have also indicated in the Discussion that LUF7346 is a type-1 activator, due to its mechanism of action (lines 276-280).

We also compared the effects of LUF7346 (7f) with two reported older modulators (ML-T531 [7a] and VU0405601 [7r]) in our *in vitro* radioligand binding assays (Yu Z et al. *Eur J Med Chem*, 2015). LUF7346 accelerated the dissociation of [³H]dofetilide to a larger extent than ML-T531 (Zhang et al. *Proc Natl Acad Sci U.S.A.*, 2012) and VU0405601 (Potet F et al. *J Biol Chem*, 2012). Furthermore, we also compared the effects of LUF7346 with two known hERG activators (NS-1643 and Rottlerin, Fig. EV4) in both LQT1^{R594Q}- and JLNS^{R594Q}-CMs. LUF7346 produced the strongest QT interval shortening. Regarding the comparison with non-allosteric modulators, hERG blockers like astemizole increased the action potential duration (Fig. 4), while LUF7346 at higher concentrations shortened the action potential duration (Fig. 3 and 4). Taken together, our findings on LUF7346 in a number of patient-derived hiPSC-CM models obviously pave the way for further drug development in correcting inherited or acquired LQTS.

2. Reversal of QT-prolongation vs. antiarrhythmic activity

One of the oldest hERG activators was NS1643 reported 10 years ago (Hansen RS et al. Mol Pharmacol. 2006 Jan;69(1):266-77). Discovery of NS 1643 evoked an enormous interest and consequently that compound was employed in sophisticated transgenic rabbit models of long QT. As described for LUF7346 in the present manuscript the old compound NS1643 could revert APD-prolongation and was successful in rescuing drug-induced long-QT syndrom (Diness TG et al. Cardiovasc Res. 2008 Jul 1;79 (1):61-9.), but increased arrhythmias in a Langendorff-model of genetic long QT (Bentzen BH et al. J Cardiovasc Pharmacol. 2011 Feb;57 (2):223-30). Collectively, this results clearly illustrates that reversal of APD and or QT-prolongation alone cannot be as a surrogate parameter to predict effective treatment of long-QT.

In the paper referred to, the older hERG activator NS-1643 rescued drug-induced LQTS by reverting the APD prolongation *in vivo*, while it increased the incidence of arrhythmias in the Langendorff experiments (Bentzen BH et al. *J Cardiovasc Pharmacol* 2011). In fact, the discrepancy between the absence of arrhythmogenic events *in vivo* and arrhythmias in isolated rabbit hearts paced at the non-physiological (for rabbits) frequency of 1 Hz, indicates a poor predictive value also for isolated hearts in these conditions.

In our study, the aim was to search for compounds with higher potency in normalizing the APD with fewer side effects originating from the shortening of APD or an excessive increase of the beating frequency, and LUF7346 displayed mitigated profiles in this regard. As arrhythmias can be regulated through different ion channels expressed in cardiomyocytes, the use of LUF7346 in LQTS should be further validated in more complicated and physiological models in the near future.

Minor points

1. QT interval under basal conditions

It is one of the strongest points of the present manuscript to show QT intervals of different cell lines including their respective controls under basal conditions. However at least from my point of view interpretation is rather tricky. In order to detect small but potentially harmful effects on QT in humans different rate algorithms were developed in order to adapt QT to actual heart rate. I would not expect that such formulations used in safety pharmacology could be simply transferred to hiPSC-CM. However, I would to expect longer QT at slower rate. LQT1 cells beats almost two times slower, but QT is only slightly increased? Furthermore, why is QT not different in LQT2N961 vs. LQT2corr?

Here, we believe we measured QT intervals for the first time under the same experimental conditions in CMs from 9 hiPSC lines (mutated and controls), allowing a direct comparison among them. The new Figure EV2 (and Table EV2) shows the analysis of the QT-RR relationships in hiPSC-CMs and correction with Bazett's and Fredericia's formulae. When we fitted our raw data with a multiple-axis regression model, the distribution of QT intervals showed angular coefficient not different from 0,

suggesting that no correction should be applied. Therefore, we agree with the referee in saying that corrections that were developed for rates within the clinical rates cannot be directly applied to hiPSC-CMs. These results might be a characteristic typical of hiPSC-CMs and may require further experiments/discussions with data from a consistent number of cell lines. We have now discussed this point (lines 294-300).

With regards to the comparison between LQT1^{R594Q}- and WT-CMs, we agree that it is not an expected result, but it can be partially explained by the QT interval correction and could eventually be another point in favour of using isogenic triplets, for which there are no examples in literature so far, also for disease modeling.

With regards to the comparison between LQT2^{N996I} and LQT2^{corr}, we apologise for erroneously omitting the asterisk indicating significant difference that is now clearly indicated in Fig. 2 and in Fig. 2 legend.

2. Findings with astemizol and personalized medicine

Authors argue the proposed measurements in hiPSC-CM could be useful as a tool to assess risk for QT-prolongation in the context of personalized medicine. I strictly disagree. From the today perspective astemizol is just an ordinary hERG blocker (Zhou Z et al. J Cardiovasc Electrophysiol. 1999 Jun;10(6):836-43). The disastrous incidence of torsade de points arrhythmias and sudden death leading to withdrawn from the market was rather the result of (at that time unknown) pharmacokinetic interaction on the level of cytochromes than peculiar pharmacodynamics that could be elucidate only by employing hiPSC-CM.

We accept that this referee might be sceptical about the value of hiPSC-CMs in contributing to the development of personalised medicine; however data are accumulating that do support the hypothesis of their potential value in multiple studies including a recent study by Burridge (Burridge et al. Nat Med, 2016) in a surprisingly small cohort.

With regards to the choice of Astemizole withdrawal from the market, this was based on experiments indicating that both Astemizole and its metabolites are known to be highly selective hERG blockers with IC50 in the nanomolar range (Zhou Z et al. Cardiovasc. Electrophysiol. 1999; Matsumoto S et al. Drug Metab. Dispos. 2002; Matsumoto S et al. Xenobiotica 2003). Similarly, Terfenadine, whose metabolite produced by cytochrome enzymes, is a potent (although less than AST) hERG blocker. Furthermore, Astemizole causes QT-prolongation. Based on these two profiles, it has been chosen by us to validate the application of hiPSC-CM model in drug screening and safety pharmacology.

3. Introduction: Trigger (p.3, l. 52)

Not only the acquired form of LQT is triggered by the factors listed? The congenital too?

We apologise for not being clear on this, but the congenital LQTS is also triggered (and of course worsened) by these factors. We have now modified the sentence that now reads: “The acquired form by contrast is triggered in healthy individuals and LQTS mutation carriers by ancillary causes such as bradycardia, electrolyte abnormalities or drugs that target cardiac ion channels non-specifically (Roden et al., Circulation, 1996; Zarebe et al., JACC, 2003; Itoh et al., Eur H J, 2015)”

4. Results: Same heterologous cells? (p. 5, l. 107)

Please clarify

We have now clarified the sentence that now reads: “The activity of five of these molecules (chemical structures in Fig 1A) was assessed by measuring their effect on the dissociation characteristics of a radioactive probe, [³H]dofetilide, from the hERG channel (Fig 1B).”

5. Results: fast and slow component (p. 6, l. 126)

What is slow and fast component? Fast activation? Fast decay? Please be precise.

We have now clarified the concept. The sentence now reads: “Both the fast (τ_{fast}) and the slow (τ_{slow}) components of I_{Kr} deactivation obtained from the fit-of-the-tail current decay were significantly increased [...]”

6. Interpretation of the amount of APD/QT-shortening by LUF7346 (p.15, l. 327)

Authors interpret different effect size in different cell lines. However a meaningful interpretation needs data about compound effects on hERG in different cell lines.

We respectfully disagree with the referee that meaningful interpretation needs data on compound effect on hERG in different cell lines. The mechanism of action of the small molecule LUF7346 has been thoroughly characterised in wild-type hERG channels (radioligand binding assays in HEK hERG cells, I_{Kr} measurements in HEK hERG, and I_{Kr} measurements in wild-type hiPSC-CMs), as was done for all previously described activators, because this is the channel that is expressed in the general population and for which it is important that the molecule is active. Testing on different LQTS genetic backgrounds is then meaningful with regards to the impact on the APD/FPD and arrhythmic phenotypes. Therefore we would like to keep the original sentence in the discussion.

7. Comparison of LUF7346 effects on APD to effect of other hERG activators (p. 13, l. 287)

Authors compare LUF7346 effects on APD to effect of other hERG activators from the literature. This seems not justified, differences could be related to different systems.

We agree with the referee’s comment. To answer this and the major point 1 we compared the effect of LUF7346 and of two known hERG activators (NS1643 and Rottlerin, Fig. EV4) in LQT1^{R594Q}- and JLNS^{R594Q}-CMs under the same experimental conditions. Lower concentrations of LUF7346 were effective in shortening the APD. However, we appreciate that more experiments should be carried out to make a direct comparison and thus we have mitigated our statement. The sentence now reads: “Importantly, the active concentration of LUF7346 that we identified is from 5 to 15 times lower than previously reported hERG activators in human Zhang et al., *PNAS*, 2014; Kang et al., *Mol Pharmacol*, 2005) or rodent Yu et al., *Circ: Arrhythm Electrophysiol*, 2016) cells, although direct comparisons should be made under identical experimental conditions.”

8. Discussion in general

The discussion would profit from a clear structure.

We agree with the referee. We have revised and reorganised the Discussion, which is now more focussed.

9. Methods: voltage clamp compensation (p. 20, l. 459)

I am wondering that authors have used capacitance and series resistance compensation only when necessary. Please clarify

We apologise for not being clear. Cell capacitance and series resistance were calculated and compensated in each cell with at least 65% of series resistance compensation. We have modified the sentence and indicated the average error on the superimposed voltage: “Cell capacitance and series resistance were calculated in each cell and compensated from 65 to 80% to maintain the error on the superimposed voltage below 5 mV (average value: 3.05 ± 0.34 mV).”

10. Proper management ("The paper explained") (p. 23, l. 525)

Authors report prevalence data of LQTS that can lead to sudden cardiac death if it is "not properly managed". This sentence is rather confusing. Do the authors already know what proper management could be?

We appreciate this comment and we have removed this statement. The sentence now reads: "The Long-QT Syndrome (LQTS) is an arrhythmogenic disorder of the heart, with a prevalence of 1:2000 newborns, which may lead to sudden cardiac death on the onset of the first arrhythmogenic event."

11. Results: Table III (p. 37, l. 894)

In HEK293 cells expressing hERG authors report results of bi-exponential fits to decaying current traces. Obviously deactivation was slowed by LUF7346 and tracings could be fitted by one time constant only. However, where is the single tau for respective mono-exponential fit at voltages >-80mV?

We apologise for not writing the value of the tau from the mono-exponential fits, which we have now indicated in Table III.

2nd Editorial Decision

13 June 2016

Thank you for the submission of your revised manuscript to EMBO Molecular Medicine. We have now received the enclosed reports from the referees that were asked to re-assess it. As you will see, while referee 1 is now globally supportive, referee 2 remains unconvinced.

I would appreciate if you could provide a point by point response to all arguments from both referees. In particular we would like you provide statistics, better graph display and state the limitations of your work as highlighted by both referees.

Please submit your revised manuscript within two weeks. I look forward to seeing a revised form of your manuscript as soon as possible.

***** Reviewer's comments *****

Referee #1 (Comments on Novelty/Model System):

The technical data quality is high but the use of a single clone per group for most of the data is an important limitation.

Referee #1 (Remarks):

Response to rebuttal for manuscript number EMM-2016-06260

The authors have provided explanations for the concerns raised and additional data as needed. Overall the manuscript is greatly improved. The fact that the study is based on a single clone of each isogenic line does undermine the work to some extent and weakens the implications of the data on the importance of using isogenic lines.

1. One concern was the demonstration that the iPSC-derived CM lines used were equivalent in both the expression of relevant genes and resulting ionic currents. The authors cite several studies related to the differentiation of hPSC-CMs and argue that different protocols only effect percentage of cTnT+ CMs rather than the quality of the cells. While this may be true, there has not been a single study that compares the resulting hPSC-CMs from different protocols in terms of electrophysiologic properties or markers of cardiomyocyte maturity. The authors do provide gene expression data of a small number of relevant ion channels but do not include any statistics for the opposed differences or explanations for some unexpected results. Namely why is the expression of SCN5A and HCN4 as high or higher in the undifferentiated iPSCs as in the differentiated hPSC-CMs? There do appear to be significant differences in the expression of KCNE2 across several lines but without appropriate statistics, it is difficult to evaluate these observed changes. Please provide appropriate statistical comparisons or separate graphs without a log axis, as this can highlight small changes which may not be statistically significant.

2. There was a concern about the low spontaneous beating rate of the cells and the significant variability between lines. To address this issue, the authors performed corrections of the measured QT intervals with respect to beating frequency using Bazett's and Fridericia's formulae. They highlight their concern for an over-correction that they feel would be misleading but did include a figure with the corrected QT intervals QTcB. This is sufficient to satisfy my concern, but the manner in which the data is displayed in figure 2B makes it difficult to see the appropriate differences because of the number of asterisk denoting statistical significance. These asterisks should be removed so that the Y-axis scale can be made more appropriate.

3. Data was asked for to specifically address possible effects of LUF7346 on the other relevant ionic channel conductances. The authors provided additional voltage clamp and action potential clamp data that satisfies that there is no significant effect of LUF7346 on the other relevant ion currents.

4. An explanation of why isogenic lines were necessary to the study. The authors provide a reasonable explanation, specifically highlighting that because of the differences observed in the QTs between lines, it would be inappropriate to use a randomly selected WT line as a control for a particular mutant line. Of course, this might be greatly mitigated if all of the lines were paced and therefore had the same beating frequency. While not explicitly stated, a more detailed analysis of the gene expression data may explain of the differences in QT observed between lines and may be an avenue of further study.

5. A specific concern was raised about the magnitude of the variation of QT between lines in comparison to different clones from the same line. Given the variance see overall, this was a concern that would make it difficult to draw conclusions about mutant-dependent changes in the QT or AP duration. The authors argue that measurements of the QT and AP duration in multiple iPSC clones is beyond the scope of the study. but do include a single example of multiple clones. Their data does suggest that the variation in AP duration between clones is minimal as compared to different lines. However, in contrast to their central argument, the clones tested were non-isogenic but have similar levels of AP prolongation.

Minor points

1. The explanation is satisfactory.
2. This inclusion in the introduction is reasonable.
3. The new graph format is an improvement.
4. This is acceptable
5. While the short-term variability clearly demonstrates the effects of AST and rescue by LUF7346, there is no comparison made with a WT line. Again, the argument for isogenic lines is understood, but it would be instructive to observe the effects of AST and LUF7346 in a line with a much higher intrinsic beating frequency. This is a minor point. The quantification of EADs is acceptable.
6. This change is noted

7. Agreed, the discussion is improved, but could be further focused on the main points of the manuscript namely the benefit of isogenic controls to compensate for variations in baseline beating frequency and QTs, the ability of LUF7346 to correct both genetic and acquired long QT and the utility of using a multi-modal approach to study the effects of a novel compound prior to further drug development. Comments about relative drug concentrations in different experimental contexts can be removed as this is well-understood.

Referee #3 (Remarks):

The conceptual weakness of the study is obvious. hERG activators are not new. To identify a new hERG activator with a higher potency no hiPSC-CM are needed. The argument of fewer side effects is not even addressed. The advantage to use hiPSC-CM over established model remains unclear. Answers to reviewers reveal more important methodological problems.

Authors have addressed many point addressed by all the reviewers; however, some relevant problems persist.

1. QT does not show rate-dependency

Authors refrain to adjust FPD to rate as suggested by reviewer 1 and 3. They demonstrate in very detail FPD not related to rate in hiPSC-CM. However, from my point of view this finding cannot be used as an argument that rate correction of QT is not necessary. The finding that hiPSC-CM do not replicate very basic physiological finding (APD shortened at higher rate) implies some profound differences in APD regulation in hiPSC-CM compared to known pharmacological models based on animal hearts. This is even more relevant since hERG channels have enormous impact on rate dependency of APD.

2. Isogenic controls and relationship between in vivo QT and APD/FPD measured in hiPSC-CM
I cannot follow the authors regarding the advantage of using isogenic controls. The finding that there is no close relationship between QT and APD/FPD question the whole concept to use APD/FPD measured in hiPSC-CM as a surrogate to predict QT-effects in humans.

2nd Revision

24 June 2016

Referee #1 (Comments on Novelty/Model System):

The technical data quality is high but the use of a single clone per group for most of the data is an important limitation.

We agree with the referee that using a single clone per group is a potential limitation for all studies using hiPSC for disease modelling even though the vast majority of literature still reports in depth characterization of only one clone per line, especially when targeted clones are used. Although this will possibly change as automatic phenotyping and genetic manipulation techniques further improve in terms of cost, time, efficiency, and ease of use, we think that the consistency of results across lines to some extent mitigates the requirement here. Nevertheless to acknowledge the limitation, we have mentioned this clearly in the discussion (page 13 – l. 296-299).

Referee #1 (Remarks):

Response to rebuttal for manuscript number EMM-2016-06260

The authors have provided explanations for the concerns raised and additional data as needed. Overall the manuscript is greatly improved. The fact that the study is based on a single clone of each isogenic line does undermine the work to some extent and weakens the implications of the data on the importance of using isogenic lines.

We thank the reviewer for appreciating the revised version of the manuscript and the new data. We agree that using a single clone per each isogenic line is a limitation of our work and more in general of the whole field. In fact gene targeting efficiencies based on the classical approach using replacement vectors are low, so that usually one clone per line is used for the targeting and typically a single targeted clone is thoroughly characterised (Tenzen et al. *J Cell Physiol* 2010; Nieminen et al., *Exp Cell Res*, 2010; Bellin et al., *EMBOJ*, 2013; Zhang et al., *PNAS*, 2014).

We have stated this limitation in the discussion (page 13 – l. 296-299).

1. One concern was the demonstration that the iPSC-derived CM lines used were equivalent in both the expression of relevant genes and resulting ionic currents. The authors cite several studies related to the differentiation of hPSC-CMs and argue that different protocols only effect percentage of cTnT+ CMs rather than the quality of the cells. While this may be true, there has not been a single study that compares the resulting hPSC-CMs from different protocols in terms of electrophysiologic properties or markers of cardiomyocyte maturity. The authors do provide gene expression data of a small number of relevant ion channels but do not include any statistics for the opposed differences or explanations for some unexpected results. Namely why is the expression of SCN5A and HCN4 as high or higher in the undifferentiated iPSCs as in the differentiated hPSC-CMs? There do appear to be significant differences in the expression of KCNE2 across several lines but without appropriate statistics, it is difficult to evaluate these observed changes. Please provide appropriate statistical comparisons or separate graphs without a log axis, as this can highlight small changes which may not be statistically significant.

We have repeated the experiment once more and we have followed the referee's suggestion providing separated graphs for each gene using a non-logarithmic scale. We have also run the statistics to identify significant differences, which are now clearly indicated in figure EV1A and its legend. The referee is correct in observing that the expression level of *SCN5A* is overall low in hiPSC-CMs, but this is not a new observation in the field (Hoekstra et al. *Frontiers Physiol*, 2012). With regards to *HCN4*, in the specific case of the undifferentiated hiPSCs we apologise for not excluding one outlier from the analysis. We agree that *HCN4* gene expression is relatively small in hiPSC-CMs, but this is in agreement with the observation that automaticity in hiPSC-CMs originates from a calcium-clock mechanism rather than from the I_f (Kim et al. *J Mol Cel Cardiol*, 2015). *KCNE2* expression is also extremely low in most of the samples, although no significant differences were observed among the different lines. Finally, *KCNJ12* was differently expressed across the lines, but no significant difference was observed within each isogenic pair. This is also a more general observation, since our data indicate that gene expression is more similar between hiPSC-CMs from isogenic lines than from independent lines.

Importantly, we would like to point out that a more meaningful analysis should take into consideration multiple levels of analysis, including protein expression and post-translational regulation, which also contribute to the shape of the action potential. As a consequence, some variability in the electrophysiological phenotype can be expected in hiPS-CMs with different genetic backgrounds. Although we think it would be useful to discuss this in the paper we think that it would make it too long to do this adequately and it would be more suited to a review. We have thus just mentioned the point in brief (page 14 – l. 305-307).

2. There was a concern about the low spontaneous beating rate of the cells and the significant variability between lines. To address this issue, the authors performed corrections of the measured QT intervals with respect to beating frequency using Bazett's and Fridericia's formulae. They highlight their concern for an over-correction that they feel would be misleading but did include a figure with the corrected QT intervals QTcB. This is sufficient to satisfy my concern, but the manner in which the data is displayed in figure 2B makes it difficult to see the

appropriate differences because of the number of asterisk denoting statistical significance. These asterisks should be removed so that the Y-axis scale can be made more appropriate.

We appreciate that Figure 2B is full, but we strongly believe that statistical significance should be clearly indicated to highlight meaningful differences. Should the editor find it useful, we can provide the figure without asterisks in the Supplementary Information.

4. An explanation of why isogenic lines were necessary to the study. The authors provide a reasonable explanation, specifically highlighting that because of the differences observed in the QTs between lines, it would be inappropriate to use a randomly selected WT line as a control for a particular mutant line. Of course, this might be greatly mitigated if all of the lines were paced and therefore had the same beating frequency. While not explicitly stated, a more detailed analysis of the gene expression data may explain of the differences in QT observed between lines and may be an avenue of further study.

We appreciate the referee's point, but we would like to highlight that while QT measurements in Figure 2 refer to non-paced hPSC-CMs, Figures 3 and 4 show action potentials of hiPSC-CMs paced at 1 Hz for the majority of the lines used in this study (LQT2^{corr}, LQT2^{N996I}, LQT1^{corr}, LQT1^{R190Q}, WT, LQT1^{R594Q}, and JLNS^{R594Q}). These data support our choice of using isogenic lines for this study. We agree with the referee that a more detailed analysis of the gene expression along with electrophysiological characterisation under pacing may indeed better explain QT differences. This would require qPCR data analysis of the same (single) cell measured by patch clamp which although challenging could indeed pave the way for further studies.

We have discussed this point on page 14, l. 305-307.

5. A specific concern was raised about the magnitude of the variation of QT between lines in comparison to different clones from the same line. Given the variance see overall, this was a concern that would make it difficult to draw conclusions about mutant-dependent changes in the QT or AP duration. The authors argue that measurements of the QT and AP duration in multiple iPSC clones is beyond the scope of the study. but do include a single example of multiple clones. Their data does suggest that the variation in AP duration between clones is minimal as compared to different lines. However, in contrast to their central argument, the clones tested were non-isogenic but have similar levels of AP prolongation.

As far as we understand, the reviewer is questioning the real value of isogenic lines, since in the example we provided (Moretti et al, *New Engl J Med*, 2010) the variation in AP duration is smaller between clones from the same line than between different lines.

With the knowledge we have now on both our own work and that of others, we believe that just finding two controls with similar QT interval may be not robust enough. We now know from independent research in our lab on endothelial differentiation (unpublished data) that, in general, inter-clone variability is relatively small, as well as inter-passage variability; conversely, interline variability is significantly larger. Therefore, with this knowledge of hindsight, we consider the use of isogenic lines as the way to move forward and reduce a source of variability due to differences in the genetic background.

Minor points

5. While the short-term variability clearly demonstrates the effects of AST and rescue by LUF7346, there is no comparison made with a WT line. Again, the argument for isogenic lines is understood, but it would be instructive to observe the effects of AST and LUF7346 in a line with a much higher intrinsic beating frequency. This is a minor point. The quantification of EADs is acceptable.

We thank the referee for this comment and accordingly we have analysed the short-term variability in the WT line, which we show in new Figure EV5.

7. Agreed, the discussion is improved, but could be further focused on the main points of the manuscript namely the benefit of isogenic controls to compensate for variations in baseline beating frequency and QTs, the ability of LUF7346 to correct both genetic and acquired long QT and the utility of using a multi-modal approach to study the effects of a novel compound prior to further drug development. Comments about relative drug concentrations in different experimental contexts can be removed as this is well-understood.

We have followed the reviewer's suggestion and removed the comments about relative drug concentrations in different experimental contexts, i.e. single cells and small clusters (pages 14-15).

Referee #3 (Remarks):

The conceptual weakness of the study is obvious. hERG activators are not new. To identify a new hERG activator with a higher potency no hiPSC-CM are needed. The argument of fewer side effects is not even addressed. The advantage to use hiPSC-CM over established model remains unclear. Answers to reviewers reveal more important methodological problems.

We appreciate the referee's criticisms on the caveats of hiPSC-CMs; however, we believe we clearly demonstrated that their predictive value in drug testing is at least as valid (or not worse) than other currently used systems, such as HEK cell ectopic expression systems, neonatal rat cardiomyocytes, and transgenic rabbits. In concordance with our view, the CiPA (Comprehensive In Vitro Proarrhythmia Assay) initiative supports and promotes validation and implementation of hiPSC-CMs for further updating the official regulatory strategy of safety pharmacology (E14 guidance).

We also appreciate the argument that hERG activators have been used before including by some of the present authors and we clearly refer to previous work in the introduction and the discussion. We believe that the pipeline presented here can be applied to multiple classes of compounds and should be compared with the poorly translatable results obtained in mice and rats in the cardiovascular field. With this approach we validated a new and effective hERG activator that could rescue genetic and acquired LQTS *in vitro*. In particular, although we used a different technical approach (patch clamp vs optical mapping), we observed a different outcome in testing LUF7244 in hiPSC-CMs (present work) versus neonatal rat CMs (Yu et al. *Circ Arrhythm Electrophysiol*, 2016). These results support the need for new and complementary approaches in the drug discovery and drug safety processes, before moving to *in vivo* experiments and to clinical trials.

Authors have addressed many point addressed by all the reviewers; however, some relevant problems persist.

1. QT does not show rate-dependency

Authors refrain to adjust FPD to rate as suggested by reviewer 1 and 3. They demonstrate in very detail FPD not related to rate in hiPSC-CM. However, from my point of view this finding cannot be used as an argument that rate correction of QT is not necessary. The finding that hiPSC-CM do not replicate very basic physiological finding (APD shortened at higher rate) implies some profound differences in APD regulation in hiPSC-CM compared to known pharmacological models based on animal hearts. This is even more relevant since hERG channels have enormous impact on rate dependency of APD.

It is a little unfortunate that the reviewer did not appear to notice the changes we made in the revised Figure 2, where we show QT corrections for all the lines tested.

We would like to highlight that the detailed analysis that we have provided to comment on rate correction formulae do not show absence of rate dependence of AP, rather that this dependence is not always linear.

To further prove that hPSC-CMs show a negative rate dependency, we provide unpublished data below on hPSC-CMs measured with patch clamp paced at three different frequencies ($n > 10$ for each data point).

[The unpublished data provided for the referees were removed]

2. Isogenic controls and relationship between in vivo QT and APD/FPD measured in hiPSC-CM

I cannot follow the authors regarding the advantage of using isogenic controls. The finding that there is no close relationship between QT and APD/FPD question the whole concept to use APD/FPD measured in hiPSC-CM as a surrogate to predict QT-effects in humans.

We respectfully disagree with the referee's comment that "no close relationship between patient's QT measured in the ECG and *in vitro* APD/FPD questions the predictive value of hiPSC-CMs". Importantly, even isolated native human ventricular myocytes (APD₉₀ = 213 ms at 0.8 Hz, Magyar et al., *Pflugers Arch*, 2000; APD₉₀ = 321 ms at 1 Hz, Maltsev et al., *Circulation*, 1998; APD₉₀ ~380 ms at 1 Hz, Coppini et al., *Circulation*, 2013) do not mirror the average QTc interval of the human population (QTc = 429 ms, Zhang et al., *Arch Int Med*, 2011; QTc = 431 ms, Straus et al., *JACC*, 2006). We believe that hPSC-CMs still harbour some predictive value, especially when analysing effects before and after drug treatment. However, we appreciate that the referee's comment could pave the way for further studies.

We have highlighted in the limitations of the study that hiPSC-CMs are immature and therefore that there are differences in ion channel expression compared to adult cardiomyocytes. Furthermore, *in vitro* monotypic cultures and single cells cannot easily reproduce the complex system of the tridimensional heart, which is composed of different cell types and cardiomyocyte subtypes, and is also exposed to hormones and environmental stimuli. We do believe however that additional predictive models, complementary to the existing small/medium animal models need to be developed, especially looking for better representation of the human physiology in terms of ion channel expression and AP features; the huge number of drugs that do not pass clinical trials after successful tests on animals is just one confirmation. Adding a platform in the preclinical phase that is more physiologically relevant than mice/rats would be beneficial for the scientific field.

In summary, we do not claim the ability of completely replacing animal models in drug discovery, but we believe that validating potential alternatives in detail might offer more relevant and complementary options.

Corresponding Author: Dr. Milena Bellin
Journal Submitted to: EMBO Molecular Medicine
Manuscript Number: EMM-2016-06260